# The Cavin-1/Caveolin-1 interaction attenuates BMP/Smad signaling in pulmonary hypertension by interfering with BMPR2/Caveolin-1 binding

Shinya Tomita[1], Naohiko Nakanishi [1✉], Takehiro Ogata[1,2], Yusuke Higuchi[1], Akira Sakamoto[1], Yumika Tsuji[1], Takaomi Suga[1] & Satoaki Matoba [1]

Caveolin-1 (CAV1) and Cavin-1 are components of caveolae, both of which interact with and influence the composition and stabilization of caveolae. CAV1 is associated with pulmonary arterial hypertension (PAH). Bone morphogenetic protein (BMP) type 2 receptor (BMPR2) is localized in caveolae associated with CAV1 and is commonly mutated in PAH. Here, we show that BMP/Smad signaling is suppressed in pulmonary microvascular endothelial cells of *CAV1* knockout mice. Moreover, hypoxia enhances the CAV1/Cavin-1 interaction but attenuates the CAV1/BMPR2 interaction and BMPR2 membrane localization in pulmonary artery endothelial cells (PAECs). Both Cavin-1 and BMPR2 are associated with the CAV1 scaffolding domain. Cavin-1 decreases BMPR2 membrane localization by inhibiting the interaction of BMPR2 with CAV1 and reduces Smad signal transduction in PAECs. Furthermore, Cavin-1 knockdown is resistant to CAV1-induced pulmonary hypertension in vivo. We demonstrate that the Cavin-1/Caveolin-1 interaction attenuates BMP/Smad signaling and is a promising target for the treatment of PAH.

[1] Department of Cardiovascular Medicine, Graduate School of Medical Science, Kyoto Prefectural University of Medicine, Kyoto 602-8566, Japan.
[2] Department of Pathology and Cell Regulation, Graduate School of Medical Science, Kyoto Prefectural University of Medicine, Kyoto 602-8566, Japan.
✉email: naka-nao@koto.kpu-m.ac.jp

Pulmonary hypertension (PH) is a progressive disease in which pulmonary vascular resistance increases due to medial and intimal thickening of the pulmonary artery micro-vessels resulting in right-sided heart failure and is associated with poor prognosis[1,2]. Advances in pulmonary vasodilator therapies such as prostacyclin analogs, prostacyclin receptor agonists, endothelin receptor antagonists, phosphodiesterase-5 inhibitors, and/or soluble guanylate cyclase stimulators have improved the prognosis of pulmonary arterial hypertension (PAH)[3]. However, the molecular mechanisms underlying PAH remain incompletely characterized, and there are no curative treatments for reversing pulmonary artery remodeling. Thus, novel therapies to improve vascular remodeling by reducing intimal thickening are required.

Several gene mutations associated with bone morphogenetic protein (BMP)/Smad signaling have been reported to be closely related to PAH[4]. Mutations in BMP receptor type 2 (BMPR2) are the most common cause of PAH. Previous reports have confirmed the presence of BMPR2 gene mutations in more than 70% of heritable PAH[5,6] and in ~20% of patients diagnosed with idiopathic PAH[7,8]. The transforming growth factor (TGF)-β/BMP superfamily plays an important role in the pathogenesis of PAH. Recent studies have reported that a ligand trap for activin, a TGF-β ligand, balances cell proliferation and inhibition and is effective in PAH treatment[9,10]. However, many carriers with BMPR2 gene mutations do not develop PAH[11], and the mechanisms underlying the roles of the TGF-β/BMP superfamily in the development of PAH have not been fully elucidated. Therefore, the mechanisms of TGF-β/BMP signaling in patients with PAH need further investigation.

BMPR2 is localized in caveolae and is associated with caveolin-1 (CAV1). Caveolae are invaginations of the cell membrane or vesicular structure attached to the cell membrane with a diameter of about 50–100 nm, in which various proteins such as receptors and ion channels are localized, and are involved in cell signaling, endocytosis, and lipid regulation[12,13]. Caveolins are components of caveolae, with both interacting with each other and influencing the composition and stabilization of caveolae. Recently, CAV1 loss-of-function mutations have been reported in human PAH patients[14] and CAV1 has been classified as a gene related to PAH[4]. CAV1 interacts with BMPR2 in vascular smooth muscle cells and plays an important role in BMPR2 transfer to the cell membrane and downstream Smad signal transduction[15]. In addition, elafin, a type of elastase, enhances the binding of CAV1 and BMPR2 in pulmonary artery endothelial cells (PAECs) and promotes downstream Smad signaling[16]. Thus, CAV1 is closely related to BMPR2 and regulates BMP/Smad signal transduction in caveolae.

Cavins are component proteins associated with caveolae containing four isoforms (Cavin-1/polymerase I and transcript release factor (PTRF), Cavin-2/serum deprivation response gene (SDPR), Cavin-3/sdr-related gene product that binds to c-kinase (SRBC), and Cavin-4/muscle-restricted coiled-coil protein (MURC)), and are also involved in PH development. Previously, we demonstrated that hypoxia-induced PH was attenuated in smooth muscle-specific Cavin-4 deficient mice via the regulation of the interaction between CAV1 and Gα13 by Cavin-4 and modulation of downstream p115RhoGEF/Rho/ROCK signaling in pulmonary artery smooth muscle cells (PASMCs)[17]. In addition, Cavin-1 knockout mice show PH with reduced expression of CAV1 and other Cavins[18]. Caveolins and Cavins form complexes in caveolae, and accumulating evidence suggests the involvement of cavins in the development of PH. However, the role of cavins and the Cavin/Caveolin system in the development of PH in association with BMP/Smad signaling is not fully understood.

Therefore, the aim of our study was to elucidate the mechanisms of PAH progression by evaluating the relationship between the cavin/caveolin system and BMP/Smad signaling and to investigate novel therapeutic targets of PAH. Here, we show that Cavin-1 and BMPR2 competitively interact with the CAV1 scaffolding domain and that these interactions modulate BMP/Smad signal transduction. Cavin-1 knockdown is resistant to CAV1-induced PH by restoring BMP/Smad signaling in PAECs. Our findings highlight the pivotal role of the cavin/caveolin system in PAH and can contribute to the development of novel therapies that enable reverse remodeling of the pulmonary artery.

## Results

**Smad 1/5/9 phosphorylation is decreased in *CAV1* knockout PAECs.** We explored the involvement of BMP/Smad signaling in CAV1-associated PAH by analyzing the BMP/Smad signaling pathway in *CAV1*-knockout (*CAV1*$^{-/-}$) mice. As previously reported, *CAV1*$^{-/-}$ mice showed reduced caveolae in PAECs and PASMCs (Fig. 1a). *CAV1*$^{-/-}$ mice developed pulmonary hypertension under normoxic conditions with elevated pressure in the right ventricle, right ventricular hypertrophy, and pulmonary vascular remodeling (Supplementary Fig. 1a–c). In whole lung tissue, phosphorylation of Smad 1/5/9 and Smad 2 was not significantly different between wild-type (WT) mice and *CAV1*$^{-/-}$ mice (Supplementary Fig. 1d). Akt phosphorylation was similar in the lungs of WT and *CAV1*$^{-/-}$ mice.

We focused on the vascular lesions by isolating pulmonary microvascular endothelial cells (PMVECs) and PASMCs from WT and *CAV1*$^{-/-}$ mice. Immunofluorescence with CD31 and αSMA verified the presence of endothelial and smooth muscle cells (Supplementary Fig. 2). Cavin-1 was significantly decreased in the PMVECs of *CAV1*$^{-/-}$ mice compared to WT mice (Fig. 1b). Although BMPR2 expression was similar between *CAV1*$^{-/-}$ and WT PMVECs, Smad 1/5/9 phosphorylation was significantly decreased in *CAV1*$^{-/-}$ PMVECs compared to WT PMVECs. There was no significant difference in the phosphorylation of Smad2 between *CAV1*$^{-/-}$ and WT PMVECs. Regarding PASMCs derived from *CAV1*$^{-/-}$ mice (Fig. 1c), Cavin-1 was also decreased in *CAV1*$^{-/-}$ PASMCs compared to that in WT PASMCs. BMPR2 expression levels were not different between WT and *CAV1*$^{-/-}$ PASMCs, and Smad 1/5/9 phosphorylation was not decreased in *CAV1*$^{-/-}$ PASMCs compared to WT PASMCs. Smad2 phosphorylation was also similar between the WT and *CAV1*$^{-/-}$ PASMCs. These results indicate that CAV1 deficiency led to downregulation of Smad 1/5/9 signaling without BMPR2 reduction in PAECs.

***CAV1* knockdown decreases BMPR2 localization at the plasma membrane and Smad 1/5/9 phosphorylation in hPAECs.** PAH pathologically presents as pulmonary arteriole remodeling via the proliferation of vascular endothelial cells, vascular smooth muscle cells, and fibroblasts. We investigated the expression of Caveolins, Cavins, and BMPR2 by cell type using previously reported single-cell RNA sequencing in mouse lung tissue[19]. *Cav1* and *Cavin-1* are ubiquitously expressed but are primarily expressed in endothelial cells. However, *Bmpr2* was mostly expressed in endothelial cells (Supplementary Fig. 3). Our results for *CAV1*$^{-/-}$ mice demonstrated that phosphorylation of Smad 1/5/9 was decreased in *CAV1*$^{-/-}$ PAMVECs compared with WT PMVECs, but not in PASMCs. Therefore, we focused on PAECs to investigate the relationship between the cavin/caveolin system and BMP/Smad signaling. We elucidated the mechanism by which CAV1 deficiency causes down-regulation of Smad 1/5/9 signaling by evaluating the TGF-β/BMP signaling pathway in human PAECs (hPAECs) using *CAV1*-specific small interfering RNA (siRNA). Similar to *CAV1*$^{-/-}$ PMVECs, CAV1-knockdown hPAECs showed decreased Smad 1/5/9 phosphorylation compared to

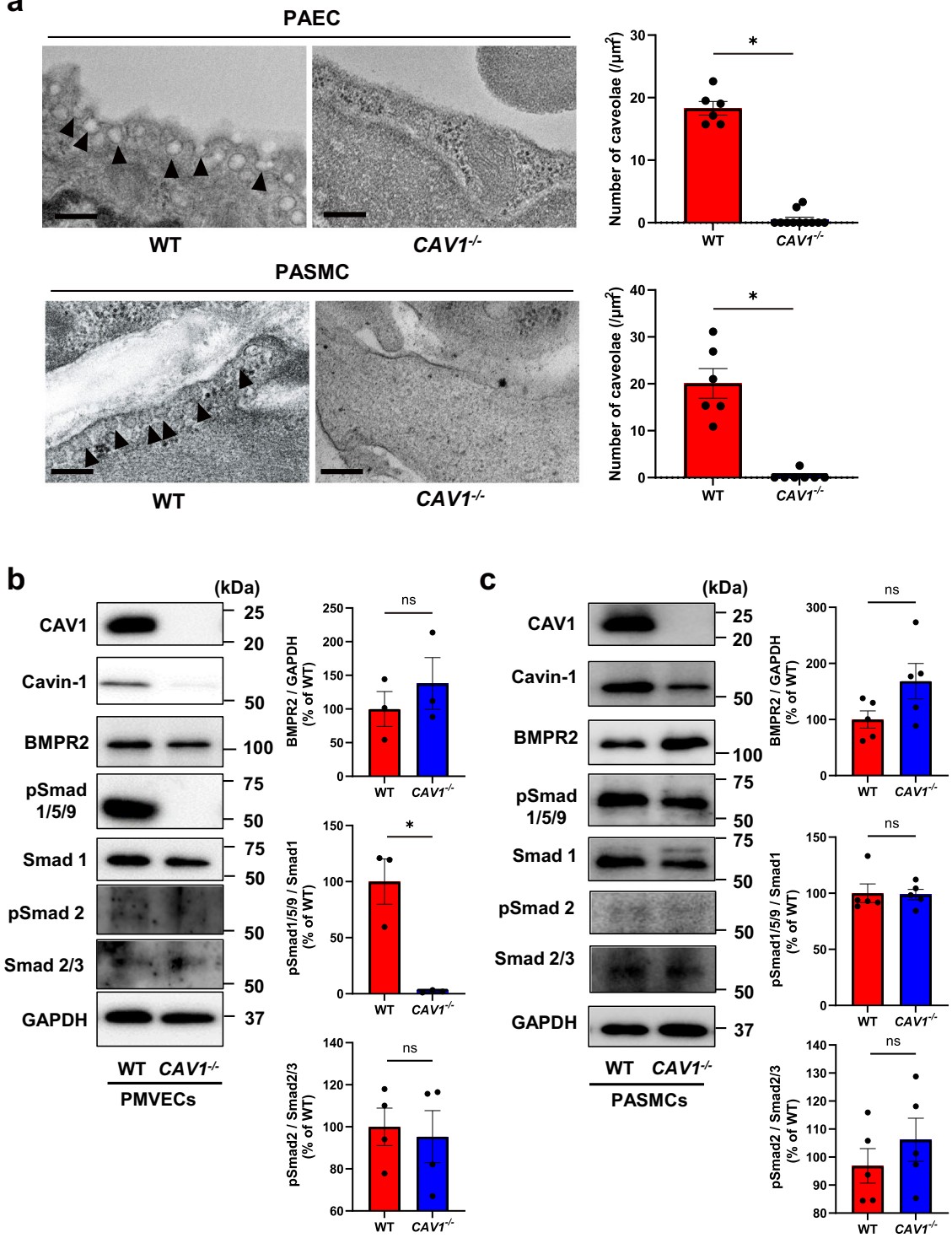

**Fig. 1 Caveolae and BMP/Smad signaling in *CAV1*<sup>−/−</sup> PAECs and PASMCs. a** Representative electron micrographs of PAECs and PASMCs in lungs from WT mice and *CAV1*<sup>−/−</sup> mice. Number of caveolae in PAECs and PASMCs of WT mice and *CAV1*<sup>−/−</sup> mice were calculated. Black arrows indicate caveolae. Scale bar, 200 nm. *$P < 0.05$ compared with WT. **b** PMVECs were isolated from the lung of WT mice and *CAV1*<sup>−/−</sup> mice. Expression of caveolae-associated proteins and BMP/Smad signaling were assessed in WT and *CAV1*<sup>−/−</sup> PMVECs. *$P < 0.05$ compared with WT. ns, not significant. **c** PASMCs were isolated from the lung of WT and *CAV1*<sup>−/−</sup> mice. Expression of caveolae-associated proteins and BMP/Smad signaling were assessed in WT and *CAV1*<sup>−/−</sup> PASMCs. ns not significant. Data are expressed as the mean ± standard error.

control hPAECs without a reduction in the total expression level of BMPR2 (Fig. 2a). Phosphorylation of Smad 2 was not affected in CAV1-knockdown hPAECs.

We next explored the localization of BMPR2 in CAV1-knockdown hPAECs. The membrane fraction by the two-step fractionation method we performed, contained the plasma membrane and several organelles: GM130 as a Golgi marker, PDI as an ER marker, and Rab7 as an endosome marker were included in the membrane fraction as well as VE-cadherin and CAV1 (Supplementary Fig. 4a). Therefore, the membrane

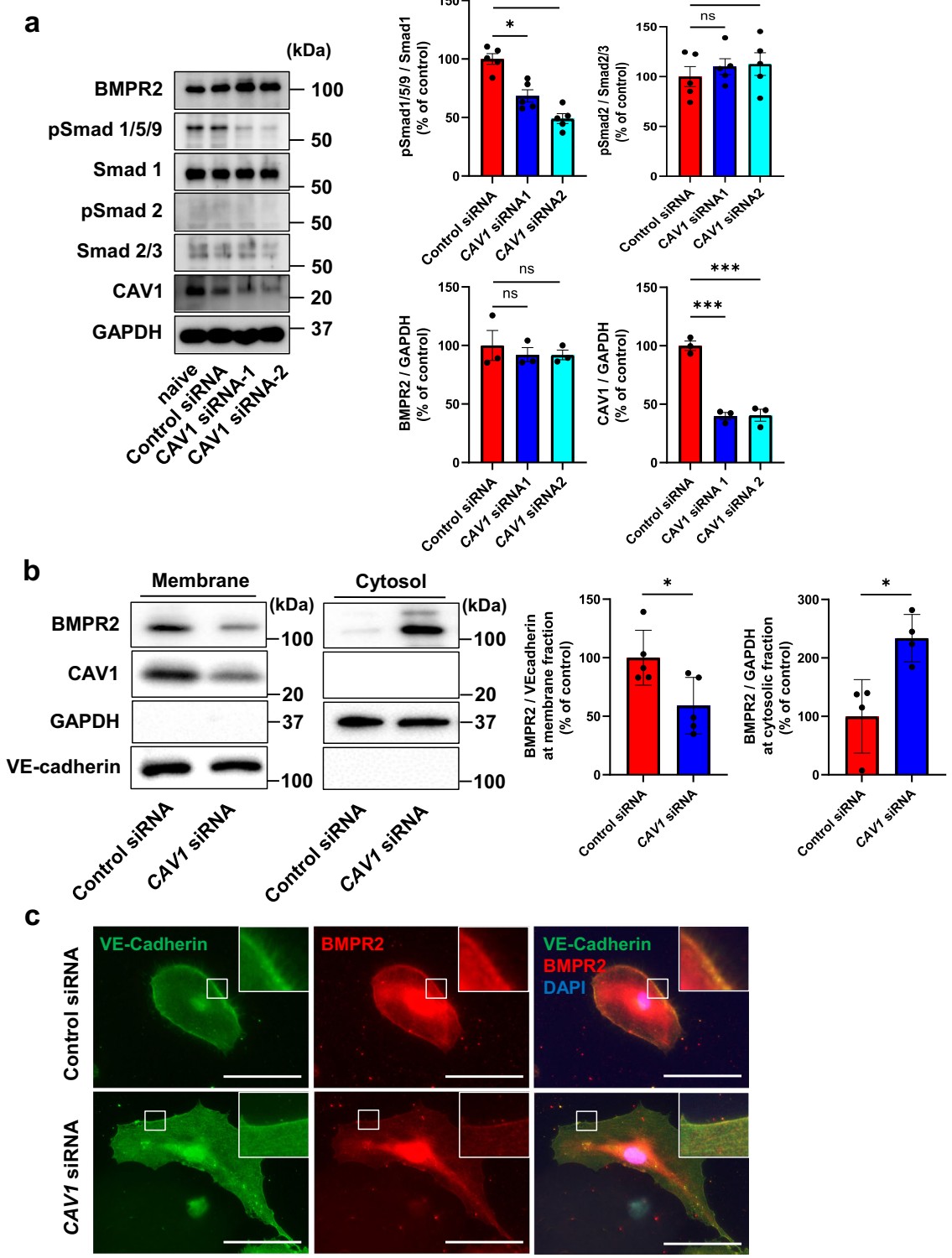

**Fig. 2 CAV1 knockdown decreases BMPR2 localization at the plasma membrane and Smad 1/5/9 phosphorylation.** hPAECs were transfected with control or *CAV1*-specific siRNA. **a** The BMP/TGF-β signaling pathway was assessed by western blotting. *$P < 0.05$ compared to control siRNA. ***$P < 0.001$ compared with control siRNA. ns, not significant. **b** Membrane and cytosol fractions were separated by sucrose gradient, and lysates of these fractions were immunoblotted. GAPDH was used as a cytosolic marker and VE-cadherin was used as a membrane marker. *$P < 0.05$ compared to control siRNA. **c** hPAECs transfected with control or *CAV1*-specific siRNA were immunostained with anti-VE-cadherin and anti-BMPR2 antibodies. Higher-magnification images are shown on the upper right. Scale bar, 100 μm. Data are expressed as the mean ± standard error.

fraction in the present study contained plasma membrane and these organelles. Most CAV1 and BMPR2 were detected in the membrane fraction of hPAECs (Fig. 2b). CAV1-knockdown decreased BMPR2 at the membrane fraction, but increased it in

the cytosolic fraction, suggesting that BMPR2 localization was altered from the plasma membrane to the cytosol in CAV1-knockdown hPAECs. Immunofluorescence revealed that BMPR2 was decreased at the plasma membrane in CAV1-knockdown

hPAECs compared to that in control hPAECs (Fig. 2c). Immunostaining showed that BMPR2 was partially co-localized with PDI, GM130, and Rab7 in PAECs. Co-localization of BMPR2 with PDI, GM130, and Rab7 was not apparently changed by CAV1 knockdown in PAECs (Supplementary Fig. 4b–d). Considering the results of Western blotting that these organelles were detected in the membrane fraction and BMPR2 was increased in the cytosol fraction in CAV1 knockdown PAECs, our results suggest that CAV1 knockdown promotes BMPR2 translocation from the membrane to the cytoplasm in PAECs. In addition, BMPR1a at the plasma membrane was also decreased in CAV1-knockdown PAECs (Supplementary Fig. 5). Consequently, these results indicate that CAV1 knockdown decreased BMPR2 localization at the plasma membrane and diminished downstream Smad 1/5/9 phosphorylation.

**Hypoxia augments Cavin-1/Caveolin-1 interaction and reduces the interaction of BMPR2 with CAV1 at plasma membrane in hPAECs.** Hypoxia plays a pivotal role in the pathogenesis of pulmonary hypertension[20,21] and induces PAECs dysfunction, apoptosis, and the activation of inflammatory and proliferative pathways[22,23]. Apoptosis-resistant proliferating PAECs are involved in the onset of pulmonary hypertension[24]. Therefore, we explored the effect of CAV1 knockdown on hypoxia-induced apoptosis in hPAECs. CAV1-knockdown hPAECs displayed resistance to hypoxia-induced apoptosis compared with control siRNA-transfected hPAECs (Supplementary Fig. 6). Next, we investigated the effect of hypoxia on CAV1 and BMPR2 localization. Although the expression and localization of CAV1 and Cavin-1 on the cell membrane were not affected by hypoxia, BMPR2 was decreased in the membrane fraction and increased in the cytoplasmic fraction of hPAECs after hypoxia stimulation (Fig. 3a). Under normoxic conditions, CAV1 and BMPR2 co-localized at the plasma membrane in hPAECs. However, BMPR2 was decreased in the cell membrane after hypoxia stimulation, whereas CAV1 levels were maintained in the membrane (Fig. 3b). The interaction of Cavin-1 with CAV1 was enhanced after hypoxia stimulation in hPAECs, as assessed by the Proximal Ligation Assay (PLA). In contrast, hypoxia inhibited the association between CAV1 and BMPR2 in hPAECs (Fig. 3c). Taken together, hypoxia augmented the interaction of Cavin-1 with CAV1, decreased the association between BMPR2 and CAV1, and decreased BMRR2 localization in the plasma membrane.

**Cavin-1 and BMPR2 competitively interact with CAV1 scaffolding domain and Cavin-1 interrupts BMP/Smad signal transduction in PAECs.** We screened for proteins that interact with CAV1 to explore the factors involved in the interaction between CAV1 and BMPR2. We performed proximity-dependent biotin identification (BioID) to identify the proteins that interact with CAV1 in hPAECs[25]. Among the screened proteins, Cavin-1 was identified using BioID (Supplementary Fig. 7). Therefore, we focused on Cavin-1 and examined its function in CAV1-associated BMP/Smad signaling in PAECs. CAV1 and Cavin-1 colocalized at the plasma membrane in hPAECs (Supplementary Fig. 8a, b). In addition, we confirmed the association of Cavin-1 with CAV1 in hPAECs using PLA (Supplementary Fig. 8c). Immunoprecipitation revealed the interaction of BMPR2 with CAV1; however, Cavin-1 overexpression decreased the association of BMPR2 with CAV1 (Fig. 4a). Although CAV1 and BMPR2 were colocalized at the plasma membrane in hPAECs by immunostaining, Cavin-1 overexpression decreased BMPR2 in the plasma membrane and decreased the colocalization of BMPR2 with CAV1 (Fig. 4b). In addition, PLA demonstrated a decreased association between BMPR2 and CAV1 in Cavin-1-

overexpressed hPAECs (Fig. 4c). Thus, Cavin-1 overexpression reduced the interaction of BMPR2 with CAV1 and BMPR2 localization to the plasma membrane in hPAECs.

Next, we attempted to identify the domain of CAV1 associated with both BMPR2 and Cavin-1 using a glutathione S-transferase (GST) pulldown assay. CAV1 contains a caveolin scaffolding domain (CSD) of 20 amino acids (residues 82–101) that binds to several receptors and signaling molecules, such as G-protein-coupled receptors, G proteins, and tyrosine kinase receptors[26–29], and regulates downstream signal transduction. Cav1(61–101), which is an oligomerization domain containing a CSD in the distal half, was associated with both Cavin-1 and BMPR2 in the GST pulldown assay (Fig. 4d). Full-length CAV1 was associated with Cavin-1 and BMPR2 in immunoprecipitation; however, CAV1 ΔCSD (81-101 deletion) did not associate with both Cavin-1 and BMPR2 (Fig. 4e), suggesting that both Cavin-1 and BMPR2 are associated with CAV1 CSD domain. In addition, the association of CAV1 (61-101) with Cavin-1 was decreased by co-expression of BMPR2, and as well as Cavin-1, the association of CAV1 (61-101) with BMPR2 was reduced by co-expression of Cavin-1 (Fig. 4f). Consistent with these results, recombinant Cavin-1 protein was found to decrease the binding of BMPR2 to CAV1 (61-101) (Fig. 4g). These results indicate that Cavin-1 and BMPR2 were competitively associated with CSD, and Cavin-1 inhibited the interaction of BMPR2 with CAV1, resulting in a decrease in BMPR2 localization at the plasma membrane.

We next assessed the function of Cavin-1 in BMP/Smad signal transduction using *Cavin-1*-specific siRNA in hPAECs. Double knockdown of CAV1 and Cavin-1 reversed BMPR2 localization in the membrane fraction, which was reduced by CAV1 knockdown (Fig. 5a). Immunostaining also revealed that Cavin-1 knockdown increased BMPR2 localization to the plasma membrane, which was reduced by CAV1 knockdown (Fig. 5b). Regarding Smad 1/5/9 signaling, CAV1 knockdown reduced the phosphorylation of Smad 1/5/9 in hPAECs; however, Cavin-1 knockdown rescued the Smad 1/5/9 phosphorylation decreased by CAV1 knockdown. Taken together, Cavin-1 interrupted BMPR2 localization to the plasma membrane by inhibiting the interaction of BMPR2 with CAV1 and suppressing BMP/Smad signal transduction in hPAECs.

*CAV1*$^{+/-}$/*Cavin-1*$^{+/-}$ **mice are resistant to CAV1-induced PH.** *Cavin-1*$^{-/-}$ mice show reduced expression of other caveolar-associated proteins, such as CAV1, Caveolin-3 (CAV3), and Cavin-2[18]. We investigated the involvement of Cavin-1 in the development of PH in vivo by generating *CAV1*$^{+/-}$ and *Cavin-1*$^{+/-}$ double heterozygous (*CAV1*$^{+/-}$/*Cavin-1*$^{+/-}$) mice. Cavin-1 expression was significantly decreased in the lungs of *CAV1*$^{-/-}$ mice (Fig. 6a and Supplementary Fig. 9). In the lungs of *CAV1*$^{+/-}$ mice, CAV1 expression decreased without affecting Cavin-1 expression, confirming the independence of CAV1 knockdown mice. In the lungs of *CAV1*$^{+/-}$/*Cavin-1*$^{+/-}$ mice, Cavin-1 expression was significantly decreased compared to *CAV1*$^{+/-}$ mice; however, CAV1 expression was comparable between *CAV1*$^{+/-}$ mice and *CAV1*$^{+/-}$/*Cavin-1*$^{+/-}$ mice. These results suggest that Cavin-1 was successfully knocked down without affecting CAV1 expression by adding *Cavin-1* heterozygous knockout.

Using WT, *CAV1*$^{+/-}$, and *CAV1*$^{+/-}$/*Cavin-1*$^{+/-}$ mice, we evaluated the number and morphology of the caveolae. Although the size of caveolae was not different among these mice, the number of caveolae was significantly decreased in *CAV1*$^{+/-}$ mice compared to WT mice. The size and number of caveolae were similar in *CAV1*$^{+/-}$/*Cavin-1*$^{+/-}$ and *CAV1*$^{+/-}$ mice, suggesting that *Cavin-1* heterozygous knockdown did not affect the number or shape of caveolae (Fig. 6b). *CAV1*$^{+/-}$ mice showed a

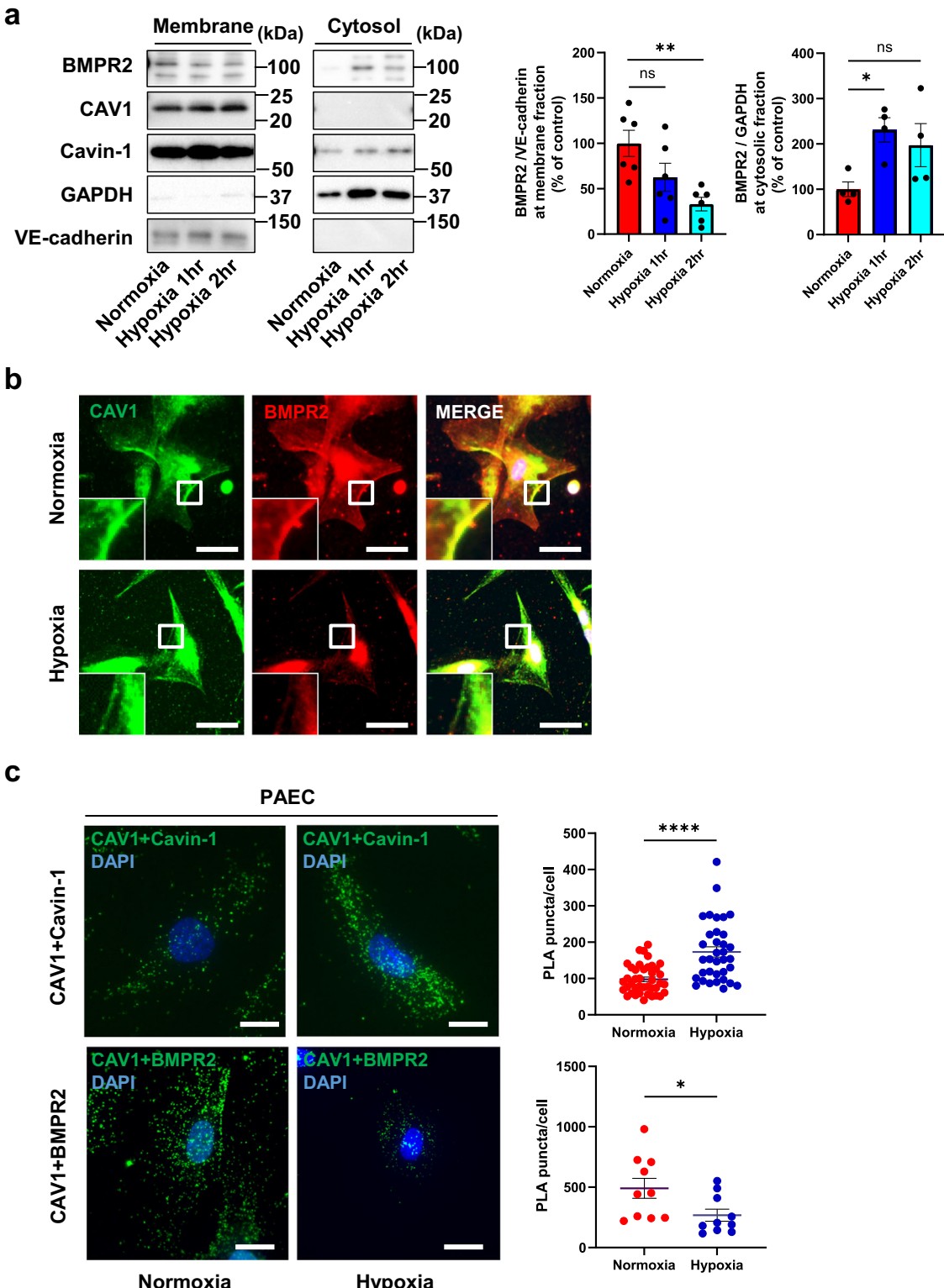

significant elevation in right ventricular systolic pressure compared to WT mice. Furthermore, *CAV1*[+/−]/*Cavin-1*[+/−] mice showed reversal of *CAV1* heterozygous-induced PH (Fig. 6c). *CAV1*[+/−]/*Cavin-1*[+/−] mice also showed improved right ventricular hypertrophy and pulmonary vascular remodeling, as observed in *CAV1*[+/−] mice (Fig. 6d, e). In addition, although Smad 1/5/9 phosphorylation was decreased in PMVECs isolated from *CAV1*[+/−] mice compared to WT mice, PMVECs isolated from *CAV1*[+/−]/*Cavin-1*[+/−] mice showed recovery of

Smad 1/5/9 phosphorylation (Fig. 6f). These results suggest that *CAV1*[+/−]/*Cavin-1*[+/−] mice are resistant to CAV1-induced PH by modulating BMP/Smad signaling in PAECs.

## Discussion

PAH is a progressive, life-threatening disease caused by pulmonary vascular remodeling[30,31]. The number of PAH patients is steadily increasing worldwide, requiring the development of effective novel therapies. Pulmonary vasodilators reduce

**Fig. 3 Hypoxia augments Cavin-1/Caveolin-1 interaction and reduces BMPR2 interaction with CAV1 and localization at plasma membrane. a** After hPAECs were transfected with control siRNA or *CAV1*-specific siRNA, hypoxia stimulation with ischemic buffer was induced for 1 or 2 h. Membrane and cytosol fractions were separated by the gradient of sucrose, and lysate of these fractions were immunoblotted. GAPDH was used as cytosol marker and VE-cadherin was used as membrane marker. *$P < 0.05$ compared with control siRNA. **$P < 0.01$ compared with control siRNA. ns, not significant.
**b** hPAECs were immunostained with anti-BMPR2 antibody and anti-CAV1 antibody after hypoxia stimulation for 2 h. Higher magnification images were shown in lower left. Scale bar, 50 μm. **c** Representative fluorescent images of the PLA for the interaction between CAV1 and Cavin-1 or CAV1 and BMPR2 in normoxia or hypoxia stimulated hPAECs. hPAECs were stimulated with normoxia or hypoxia for 2 hr and PLA were assessed with anti-CAV1 antibody, anti-Cavin-1 antibody and anti-BMPR2 antibody. The number of cell samples were 41 in CAV1/Cavin-1 normoxia, 35 in CAV1/Cavin-1 hypoxia, and 10 each in CAV1/BMPR2 normoxia and hypoxia, respectively. *$P < 0.05$ compared with normoxia. ****$P < 0.0001$ compared with normoxia. ns, not significant. Scale bar, 20 μm. Data are expressed as the mean ± standard error.

pulmonary artery pressure and prolong the prognosis of PAH patients[2]. Recently, initial upfront combination therapy with pulmonary vasodilators has been recommended for high-risk PAH patients[3,32]. Early pulmonary vascular remodeling has been shown to reverse remodeling by unloading the pulmonary arterial pressure[33], but advanced pulmonary vasculopathy, such as plexiform lesions, is fundamentally irreversible despite reducing pulmonary artery pressure. Therefore, new treatments that actively improve pulmonary artery remodeling are urgently required. Our results demonstrate that Cavin-1, a major component protein of the Cavin family, competitively inhibits the binding of CAV1 CSD and BMPR2, interrupts BMPR2 localization at the plasma membrane, and reduces BMP/Smad signaling in PAECs. Cavin-1 knockdown reversed pulmonary hypertension and vascular remodeling induced by CAV1 knockdown. These results indicate that the inhibition of the CAV1 and Cavin-1 interaction may be a new therapeutic target for pulmonary hypertension (Fig. 7).

Cavin-1 knockout mice display pulmonary hypertension[18]. At first glance, this seems to contradict our results. However, caveolin and cavin expression depends on the presence of caveolae. Cavin-1 expression was significantly decreased in the lungs of CAV1 knockout mice (Fig. 6a and Supplementary Fig. 9). Similarly, CAV1 expression was significantly reduced in Cavin-1 deficiency lungs[18]. Thus, because the expression of caveolin and cavin influence each other, it is difficult to separately assess the roles of CAV1 and Cavin-1 in PH. To resolve this problem, we generated $CAV1^{+/-}$ and $CAV1^{+/-}/Cavin-1^{+/-}$ mice. Although both $CAV1^{+/-}$ mice and $CAV1^{+/-}/Cavin-1^{+/-}$ mice had a reduced number of caveolae in PAECs compared with WT mice, there were no significant differences in caveolae size and number between $CAV1^{+/-}$ mice and $CAV1^{+/-}/Cavin-1^{+/-}$ mice. CAV1 expression was significantly suppressed in the lungs of $CAV1^{+/-}$ mice compared with that in WT mice. However, Cavin-1 expression did not decrease in the lungs of $CAV1^{+/-}$ mice, suggesting that $CAV1^{+/-}$ mice demonstrate independent CAV1 knockdown. In contrast, $CAV1^{+/-}/Cavin-1^{+/-}$ mice showed reduced Cavin-1 expression without a decrease in CAV1 expression compared to $CAV1^{+/-}$ mice, suggesting independent Cavin-1 knockdown. Therefore, these mice revealed the effect of CAV1 and Cavin-1 independently in PH and demonstrated that Cavin-1 knockdown was resistant to PH induced by CAV1 knockdown.

In general, caveolins form caveolae together with cavins. However, several cells, such as hepatocytes, neurons, and lymphocytes, express CAV1 without forming distinguishable caveolae. Non-caveolar caveolins also organize scaffolds that interact with proteins and play the roles of lipids transportation or signal transduction[34–36]. In addition, it has been reported that non-caveolar CAV1 expression promotes lymphangiogenesis[37], and Cavin-1 neutralizes non-caveolar CAV1 microdomains and inhibits CAV1 signaling function in prostate cancer[38–40]. Therefore, an alternative hypothesis is that BMPR2 interacts not

with caveolae but with non-caveolar scaffolds and that Cavin-1 induction of caveolae reduces the pool of available scaffolds for interaction. However, the number of caveolae in PAECs of $CAV1^{+/-}/Cavin-1^{+/-}$ mice was not decreased compared with that of $CAV1^{+/-}$ mice. This result suggests, but insufficiently, that Cavin-1 does not affect the scaffolding of non-caveolar CAV1. The localization of BMP receptors is also essential for BMP signaling. A part of BMPR2 mutants misdirect the localization of BMP receptors on the plasma membrane and decrease BMP-dependent Smad signaling[41]. In addition, the localization of BMP receptors in caveolae are important for activating Smad signaling[42]. In the present study, Cavin-1 decreased BMPR2 localization on the plasma membrane, and the knockdown of Cavin-1 rescued its localization to the plasma membrane and the Smad 1/5/9 phosphorylation which were reduced by CAV1 knockdown. These findings suggest that Cavin-1 regulates the Smad signaling through the BMPR2 localization on the plasma membrane.

CSD, a hydrophobic 20-amino acid sequence (residues 82–101) in CAV1, directly interacts with many proteins, such as growth factor receptor[43], insulin receptor[44], G proteins[45], eNOS[28,46], and TGF receptor[47], resulting in usually suppression of these signaling transductions to hypoactivity. Previously, we identified that Cavin-4/MURC interacts with CSD and competitively inhibits the association of Gα13 with CSD, leading to the dissociation of Gα13 from CSD to convert the Gα13 subunit from an inactive to an active form[17]. Recently, CAV1 scaffolding domain peptides (CSP) and its seven–amino acid deletion fragment peptides (CSP7) have emerged as potential treatments for several disease models[48–50]. CSP/CSP7 demonstrated anti-fibrosis effects in a lung fibrosis model by inhibiting alveolar epithelial progenitor type II cell apoptosis and fibrotic lung fibroblasts activation[51]. In addition, CSP has been shown to attenuate liver fibrosis by inhibiting TGF-β1/Smad signaling[52]. Our results showed that Cavin-1 and BMPR2 were competitively associated with CSD. Cavin-1 inhibited the interaction of BMPR2 with CSD and reduced the localization of BMPR2 from the plasma membrane, leading to the suppression of Smad 1/5/9 signaling in PAECs. These results suggest that CSP may be a potential treatment for PH by mimicking the interaction of Cavin-1 with CSD, resulting in the restoration of BMP/Smad signaling. Future studies should investigate whether CSP improves pulmonary artery remodeling and is useful in the treatment of pulmonary hypertension.

An imbalance in TGF-β and BMP signaling is implicated in the development of PAH. Several approaches have recently emerged that directly modify the TGF-β/BMP signaling pathway. FK506 (tacrolimus) interacts with FKBP12, activates Smad1/5/8 with or without BMPR2 mutations, recovers endothelial dysfunction, and ameliorates PH[53–55]. The activin receptor type IIA-Fc (ActRIIA-Fc) fusion protein (sotatercept) has been shown to restore the balance between TGF-β and BMP signaling, suppress inflammation, and reduce pulmonary vascular resistance in patients with PAH[9,10,56]. In addition, the selective activation of

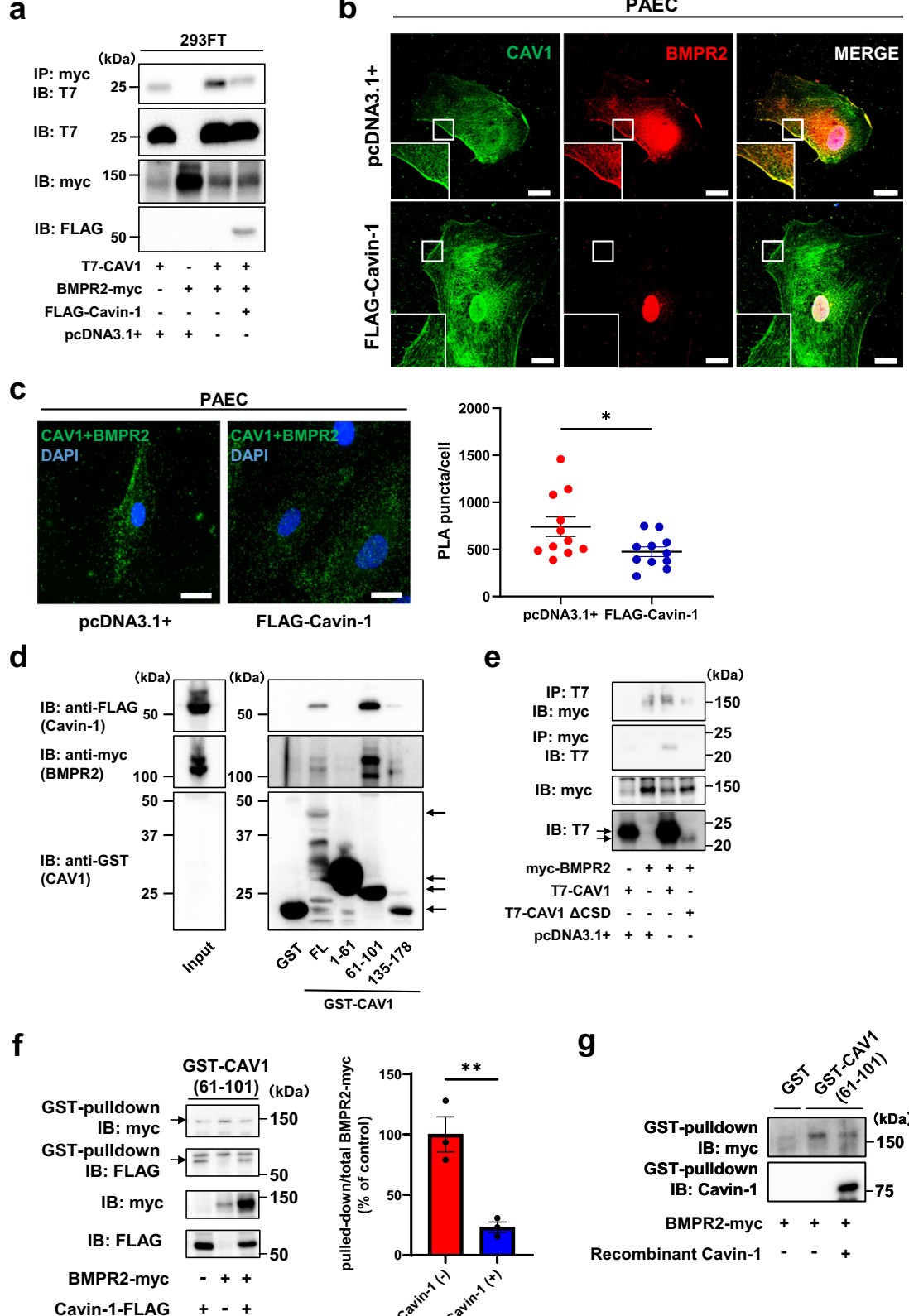

endothelial BMPR2 by BMP9 has been shown to reverse PAH[57]. Modulating TGF-β/BMP signaling can cause reverse remodeling of the pulmonary artery, which could be a curative treatment for PAH. Our study demonstrated that Cavin-1 modulated the interaction of BMPR2 with CAV1, and Cavin-1 knockdown reversed PH by restoring BMPR2 localization and Smad 1/5/

9 signaling in PAEC. In addition, *Cavin-1* knockdown reversed pulmonary vascular remodeling and PH induced by CAV1 knockdown by restoring Smad 1/5/9 signaling in PAECs. These results suggest that the Cavin-1/Caveolin-1 interaction interrupts BMP/Smad signaling in PH. Although the involvement of the Cavin/Caveolin system in the TGF-β/BMP signaling pathway has

**Fig. 4 Cavin-1 and BMPR2 are competitively associated with CAV1 scaffolding domain. a** 293FT cells were transfected with pcDNA3.1-T7-hCAV1 and/or pcDNA3-hBMPR2-myc and/or pFLAG-CMV-4-hCavin-1 and cell lysates were co-immunoprecipitated with anti-myc antibody. **b** hPAECs was transfected with pcDNA3.1+ or pFLAG-CMV-4-hCavin-1, and immunostained with anti-CAV1 antibody and anti-BMPR2 antibody. Higher magnification images were shown in lower left. Scale bar, 20 μm. **c** Representative fluorescent images of the PLA for the interaction between CAV1 and BMPR2 in pcDNA+ or pFLAG-CMV-4-hCavin-1 transfected hPAECs. The number of cell samples were 11 each in pcDNA+ or pFLAG-CMV-4-hCavin-1 transfected hPAECs. *P < 0.05 compared with pcDNA3.1+. Scale bar, 20 μm. **d** Each domain of CAV1 (FL: full length, 1–61: C-terminal domain, 61–101: oligomerization domain containing caveolin scaffolding domain (82–101), 135–178: N-terminal domain) tagged with GST was conjugated to glutathione-Sepharose beads, and incubated with lysates from 293FT cells transfected with pFLAG-CMV-4-hCavin-1 or pcDNA3-hBMPR2-myc. GST pulldown was performed with 293FT cell lysates transfected with plasmids expressing the indicated proteins. Pulled-down protein was immunoblotted with anti-FLAG or anti-myc antibodies. Arrows indicate each GST-fusion protein. **e** 293FT cells were transfected with pcDNA3-hBMPR2-myc and/or pcDNA3.1-T7-hCAV1 and/or pcDNA3.1-T7-hCAV1ΔCSD and cell lysates were co-immunoprecipitated with anti-T7 or anti-myc antibodies. **f** CAV1 (61–101) tagged with GST was conjugated to glutathione-Sepharose beads and incubated with lysates from 293FT cells transfected with pcDNA3-hBMPR2-myc and/or pFLAG-CMV-4-hCavin-1. GST pulldown was performed with 293FT cell lysates transfected with plasmids expressing the indicated proteins. Pulled-down protein was immunoblotted with anti-myc or anti-FLAG antibodies. **P < 0.01 compared with pulled-down BMPR2-myc without Cavin-1. **g** GST or CAV1 (61–101) tagged with GST were conjugated to glutathione-Sepharose beads and incubated with lysates from 293FT cells transfected with pcDNA3-hBMPR2-myc and/or recombinant Cavin-1 protein. GST pulldown was performed and pulled-down protein was immunoblotted with anti-myc or anti-Cavin-1 antibodies. Data are expressed as the mean ± standard error.

not been demonstrated in human PAH, our findings provide a crucial role for the Cavin/Caveolin system in the BMP/Smad signaling pathway in PAH development.

In conclusion, our study demonstrated that the Cavin-1/Caveolin-1 interaction attenuated BMP/Smad signaling by interfering with BMPR2/Caveolin-1 binding and was involved in the development of PH. Downregulation of Cavin-1 ameliorated CAV1 knockdown-induced pulmonary hypertension by recovering Smad 1/5/9 phosphorylation in PAECs. Our findings elucidated a previously unidentified mechanism of the Cavin/Caveolin system on BMP/Smad signaling and could be a new therapeutic target for PAH.

## Methods

**Reagents.** The rabbit polyclonal anti-CAV1 antibody (#sc-894), the mouse monoclonal anti-CAV1 antibody (#sc-70516), the monoclonal anti-BMPR1a antibody (#sc-518037) and the mouse monoclonal anti-VE cadherin antibody (#sc-9989) were purchased from Santa Cruz Biotechnology (Dallas, TX, USA); the rabbit polyclonal antibody to Cavin-1 (#ab76919) and αSMA (#ab124964), the horseradish peroxidase-conjugated monoclonal antibody to GAPDH (#ab105428) and the mouse monoclonal antibody to myc tag (#ab32) were from Abcam PLC (Cambridge, UK); the rabbit polyclonal antibodies to Akt (#9272), phospho-Akt (Ser473) (#9271), phospho-Smad1 (Ser463/465)/Smad5 (Ser463/465)/Smad9 (Ser465/467) (#13820), Smad1 (#9743), phospho-Smad2 (Ser465/467) (#3108), Smad2/3 (#3102) and the horseradish peroxidase-conjugated secondary antibodies (anti-mouse-HRP, and anti-rabbit-HRP) were purchased from Cell Signaling Technology (Danvers, MA, USA); rabbit polyclonal antibody to Cavin-2 (#12339-1-AP) and Cavin-3 (PRKCDBP) (#16250-1-AP) was purchased from ProteinTech Group, Inc. (Rosemont, IL, USA); the rabbit polyclonal anti-Cavin-4 antibody was generated as previously described[58,59]; the mouse monoclonal antibody to FLAG (clone M2, #F3165,), the mouse monoclonal antibody to T7 (#69522), Cy3-conjugated monoclonal antibody to αSMA (#C6198) and collagenase type 1 from *Clostridium histolyticum* were from Merck (Darmstadt, Germany); the mouse monoclonal antibodies to CAV3 (#610421) and the rat monoclonal antibodies to CD31 (clone MEC13.3, #550274) were from BD Biosciences (Franklin Lakes, NJ, USA); the horseradish peroxidase-conjugated monoclonal antibody to GST (#NB600-388) was from Wako Pure Chemical Industries (Osaka, Japan). The mouse monoclonal antibody to BMPR2 (3F6F8, #MA5-15827)[60], human *CAV1* and *Cavin-1*-specific, control small interfering RNA (siRNA) duplex oligonucleotides

(Silencer Select siRNAs) and Lipofectamine RNAiMAX reagent were obtained from Thermo Fisher Scientific Inc. (Waltham, MA, USA). The Duolink II Detection Kit was purchased from Sigma Chemical Co. (St. Louis, MO, USA). The Caspase-Glo 3/7 assay was purchased from Promega Co. (Madison, WI, USA). Dulbecco's modified Eagle's medium (DMEM), penicillin/streptomycin solution (×100), phosphate-buffered saline (PBS), and 4% paraformaldehyde (PFA) were purchased from FUJIFILM Wako Pure Chemical Corporation (Tokyo, Japan). DAPI Fluoromount was purchased from Southern Biotech (Birmingham, AL). hPAECs were purchased from Lonza (Walkersville, MD, USA). Recombinant GST-Cavin-1 was purchased from Abnova (Taipei)[61].

**Isolation and culture of mouse PMVECs.** Mouse PMVEC were prepared as previously described with modifications[62–65]. C57BL/6 J mice, aged 8–12 weeks, were used for PMVECs isolation. After transcardial perfusion with ice-cold PBS, the lungs were shredded and digested with Collagenase I (2 mg/mL collagenase I in serum-free DMEM) in shaking incubator (100 rpm) at 37 °C for 30 min. The minced tissue was filtered through a 70-μm cell strainer. Cell suspensions were mixed with serum to stop digestion and centrifuged at 300 g. Cell pellets were resuspended in MACS buffer (2% FBS and 2 mM EDTA in PBS (-), pH 7.20). An appropriate volume of CD31 MicroBeads (Miltenyi Biotec Inc., CA, USA) was added to the suspension, according to the manufacturer's instructions. The cells were incubated for 15 min at 4 °C and resuspended in MACS Buffer. The suspension was then applied to the prepared LS column. The LS column was washed 3 times with Wash Buffer. Subsequently, we flushed out the fraction containing the magnetically labeled cells by firmly applying the plunger. Cell suspension was centrifuged at 300 g and the pellet was cultured in endothelial basal media-2 (EBM-2) supplemented with growth factors and 2% FBS (Lonza). Mouse PMVECs were maintained at 37 °C in a humidified 5% $CO_2$ chamber, changing the media every 2 days. Cells were used for experiments without further passage upon reaching 80–90% confluent.

**Isolation and culture of mouse PASMCs.** Mouse PASMCs were isolated as previously described with modification[66,67]. C57BL/6 J mice, aged 8–12 weeks, were used for PASMC isolation. After transcardial perfusion with ice-cold PBS, 3–5 ml PA agarose (0.05 g agarose plus 0.05 g iron particles in 10 ml serum-free DMEM) was slowly injected into the right ventricle until the lungs turned gray. The lung tissue was shredded and transferred

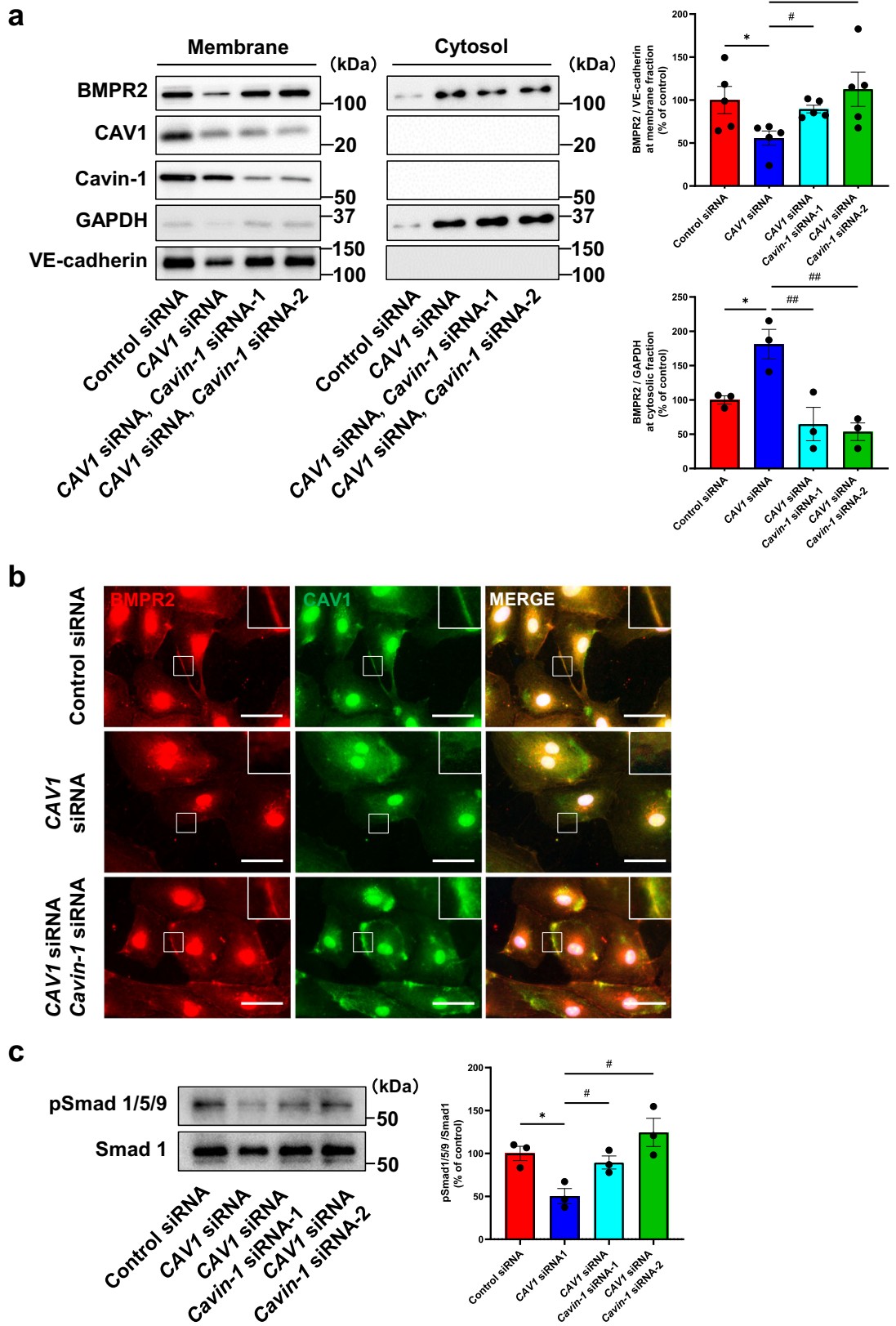

to a 50 ml centrifuge tube. Ferrous tissue was pulled down to the wall of the centrifuge tube using a magnet. After washing with PBS, collagenase solution (80 U/ml collagenase in serum-free DMEM) was added to the centrifuge tube, poured into a 60 mm culture dish, and digested for 1 h at 37 °C. Digested tissue was pulled down with a magnet and washed with complete medium

(20% FBS and 1% Penicillin/Streptomycin in DMEM) to inactivate the collagenase. Finally, we added complete medium and then the cell suspension was cultured in 35 mm culture dish (5% $CO_2$, 37 °C). Approximately 50% of the cells that first crawled out of the iron-containing vessels were smooth muscle cells, and the other 50% were fibroblasts. To obtain pure PASMC, the pulldown

**Fig. 5 Cavin-1 knockdown rescue BMPR2 localization at plasma membrane and Smad 1/5/9 phosphorylation decreased by CAV1 knockdown.**
**a** hPAECs were transfected with control siRNA and/or *CAV1*-specific siRNA and/or *Cavin-1*-specific siRNA. Membrane and cytosol fractions were separated by the gradient of sucrose. Lysate of these fractions were immunoblotted to evaluate BMPR2 localization at plasma membrane. GAPDH was used as cytosol marker and VE-cadherin was used as membrane marker. *$P < 0.05$ compared with control siRNA, #$P < 0.05$ compared with *CAV1* siRNA, ##$P < 0.01$ compared with *CAV1* siRNA. **b** hPAECs were transfected with control siRNA and/or *CAV1*-specific siRNA and/or *Cavin-1*-specific siRNA, and immunostained with anti-BMPR2 antibody and anti-CAV1 antibody. Higher magnification images were shown in upper right. Scale bar, 50 μm.
**c** Phosphorylation of Smad 1/5/9 was evaluated in hPAECs transfected with control siRNA and/or *CAV1*-specific siRNA and/or *Cavin-1*-specific siRNA. *$P < 0.05$ compared with control siRNA, #$P < 0.05$ compared with *CAV1* siRNA. Data are expressed as the mean ± standard error.

procedure with a magnet was repeated several times. Cells were cultured to the desired number after several passages.

**Immunostaining**. Specimens from the lung were fixed in 4% PFA and stained with the Cy3-conjugated αSMA antibody (1:200), and nuclei were visualized using DAPI Fluoromount. Human PAECs, C2C12 cells, mouse PMVECs, and mouse PASMCs were fixed with 4% PFA on ice for 15 min and stained with the CAV1 (1:100), Cavin-1 (1:100), BMPR2 (1:50), BMPR1a (1:50), FLAG (1:100), CD31(1:50), VE-cadherin (1:50), or αSMA (1:200) antibodies. Secondary antibodies were conjugated with Alexa Fluor 488 or 555 (1:500), and nuclei were visualized using DAPI Fluoromount. Fluorescent signals were detected using a Keyence BZ-X700 digital microscope (Osaka, Japan) or Zeiss LSM510 META Confocal Imaging System (Oberkochen, Germany).

**Plasmid constructs**. The human CAV1-expressing vector, pcDNA3.1-T7-hCAV1, and the human Cavin-1-expressing vector, pFLAG-CMV-4-hCavin-1, were gifts from Yukiko K. Hayashi (Tokyo Medical University, Tokyo, Japan)[68]. pGEX-4T-1-GST-CAV1(FL), GST-CAV1(1-61), GST-CAV1(61-101), and GST-CAV1(135–178) were gifts from William C. Sessa (Addgene plasmids #22445, 22446, 22447, and 22448)[69]. Myc-tagged pcDNA3-BMPR2 was a kind gift from Petra Knaus (Frei University Berlin, Berlin, Germany). The pcDNA3.1 plasmid was purchased from Thermo Fisher Scientific Inc. (Waltham, MA, USA). pcDNA3.1-T7-hCAV1 ΔCSD (81-101 deletion) was generated by PCR with pcDNA3.1-T7-hCAV1 using a forward primer (5′-TTGCTGTCTGCCCTCTTTGG-3′) and a reverse primer (5′-ACTGTGTGTCCCTTCTGGTT-3′).

**Immunoprecipitation**. 293FT cells transfected with each of the plasmid using FuGENE6 (Roche Applied Science, Upper Bavaria, Germany) were washed with ice-cold PBS and lysed with lysis buffer (50 mM Tris-HCl, pH 8.0, 50 mM NaCl, 1% Nonidet P-40) containing protease inhibitor cocktail (Pierce), 1 mM $Na_3VO_4$, 1 mM NaF, 1 mM phenylmethylsulfonyl fluoride, and 60 mM octyl glucoside. Immunoprecipitation was carried out by incubating the same amount of cell lysates with magnetic beads (Magnosphere MS300/Carboxyl, COSMO BIO, Tokyo, Japan) coated with each antibody at 4 °C overnight. Beads were washed with wash buffer (50 mM Tris, pH 8.0, 50 mM NaCl, 1.0% Nonidet P-40, 1 mM NaF) five times and the precipitated proteins were separated by SDS–PAGE, transferred to a polyvinylidene difluoride membrane, and probed with each antibody.

**Western blotting**. Cell lysates were extracted using radio-immunoprecipitation assay (RIPA) lysis buffer (FUJIFILM Wako) with a protease-phosphatase inhibitor cocktail. To separate the membrane fraction and cytosol fraction, hPAECs were harvested with PBS and resuspended with phosphate buffer (pH 6.8, 0.1 M $NaH_2PO_4$, 0.1 M $Na_2HPO_4$, 8.5% sucrose, protease inhibitor cocktail, 1 mM $Na_3VO_4$, and 1 mM NaF). Cells were homogenized and centrifuged at 1000×*g* at 4 °C for 10 min.

Supernatants were ultracentrifuged at 55,000 rpm at 4 °C for 30 min. Pellets were lysed and loaded as the membrane fraction and the supernatant as the cytosolic fraction. Total or fractionated cell lysates were electrophoresed in SDS sulfate-polyacrylamide PAGE and transferred to polyvinylidene difluoride membranes. Horseradish peroxidase-conjugated anti-rabbit and anti-mouse IgG antibodies were used as the secondary antibodies.

**Proximity ligation assay**. The interaction between proteins was assessed using the Duolink II Detection Kit (Sigma-Aldrich), according to the manufacturer's specifications. The signal was visualized as a fluorescent spot and captured using a Keyence BZ-X700 digital microscope. The PLA signals in each cell were counted.

**Animals**. CAV1-knockout (CAV1$^{-/-}$) mice (C57BL/6 J background) and Cavin-1-knockout (Cavin-1$^{-/-}$) mice (C57BL/6 J background) were purchased from the Jackson Laboratory. Male mice aged 8–16 weeks were used in this study. All animal care and experimentation procedures performed in this study were approved by the Institutional Animal Care and Use Committee of Kyoto Prefectural University of Medicine. We have complied with all relevant ethical regulations for animal use.

**Measurement of RV pressure and histological analyses**. The mice were intubated with a 22-gauge Teflon tube and placed in the supine position. To measure RV hemodynamics, open-chest RV catheterization using a 1.2-F pressure catheter (Transonic Scisense, Inc., London, ON, Canada) was performed under anesthesia with 1.5% isoflurane. Pulmonary vascular remodeling was assessed by measuring the medial thickness of alveolar/distal pulmonary vessels 25–100 mm in diameter from lung sections immunostained with αSMA. Percent wall thickness was expressed as the medial wall area (the area between the internal and external lamina) divided by the area of the vessel (the area between the external lamina). Multiple lung sections were prepared for each mouse, and > 5 vessels were analyzed in each lung section.

**Transmission electron microscopy and quantitation**. Twelve-week-old mouse lungs were fixed with 2% glutaraldehyde in 0.1 M cacodylate buffer, post-fixed with 2% $OsO_4$, and stained with uranyl acetate and lead citrate. Microtome sections were examined under an H-7100 transmission electron microscope (HITACHI, Tokyo, Japan) and photographed at a magnification of ×60,000. Caveolae are identified by their characteristic flask shape and location at or near the plasma membrane[70,71]. At least ten independent fields were quantitated for each condition to measure the caveolar number and perimeter.

**Gene silencing through RNA**. Human *CAV1*-, *Cavin-1*-specific, and control siRNAs were transiently transfected into hPAECs using Lipofectamine RNAiMAX reagent according to the manufacturer's protocol. The medium was changed 4 h post-transfection, and the cells were used for the assay 72 h after transfection.

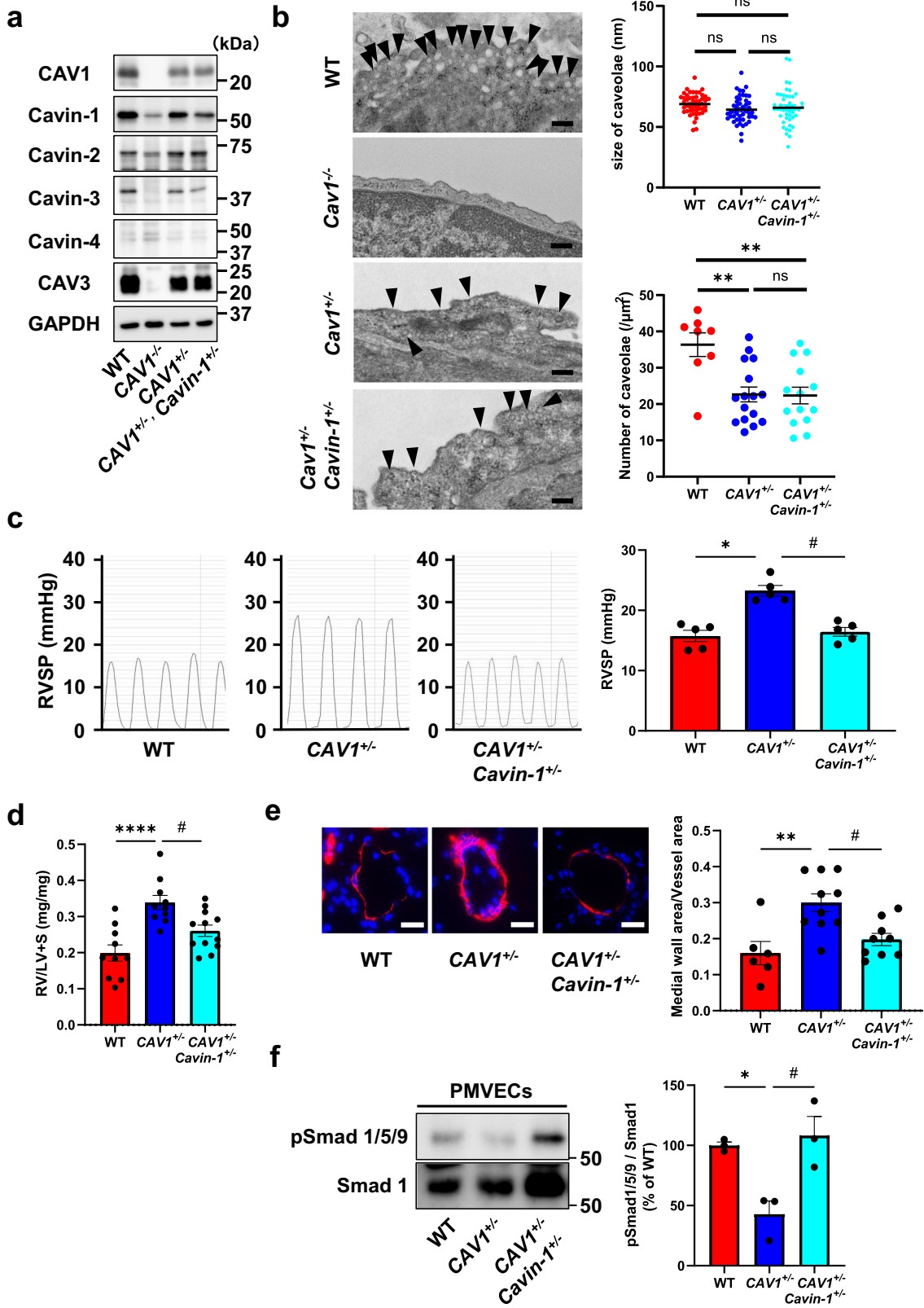

**Apoptosis assay**. hPAECs apoptosis was evaluated using the Caspase-Glo 3/7 assay (Promega). After 72 h of *CAV1*-knockdown, 5000 cells/well were seeded in 96 well plates. hPAECs were challenged with hypoxia stimulation (1% $O_2$) with modified Esumi ischemic buffer (137 mM NaCl, 12 mM KCl, 0.49 mM MgCl$_2$, 0.9 mM CaCl·2H$_2$O, 4 mM HEPES, 10 mM deoxyglucose, and 20 mM sodium lactate; pH 6.2)[72] for 2 h. Caspase activity was measured after adding 100 µl of caspase 3/7 luciferase reagent mix for 1 h as total luminescence using a TECAN microplate reader (Zurich, Switzerland).

**GST pulldown assay**. The GST pulldown assay using GST-conjugated CAV1 fragment proteins was carried out as previously

**Fig. 6 *CAV1* and *Cavin-1* double knockdown mice reverse pulmonary hypertension induced by *CAV1* knockdown. a** Caveolae-related proteins in the lung of WT, *CAV1⁻/⁻*, *CAV1⁺/⁻*, and *CAV1⁺/⁻/Cavin-1⁺/⁻* mice were assessed by Western blotting. **b** Representative electron micrographs of PAECs in lungs from WT mice, *CAV1⁻/⁻* mice, *CAV1⁺/⁻* mice, and *CAV1⁺/⁻/Cavin-1⁺/⁻* mice. Size and number of caveolae in PAECs of WT mice, *CAV1⁺/⁻* mice, and *CAV1⁺/⁻/Cavin-1⁺/⁻* mice were calculated. Black arrows indicate caveolae. *$P < 0.05$ compared with WT, **$P < 0.01$ compared with WT. ns, not significant. Scale bar, 200 nm. **c** RV systolic pressure (RVSP) in WT mice, *CAV1⁺/⁻* mice, and *CAV1⁺/⁻/Cavin-1⁺/⁻* mice under normoxic conditions were measured. *$P < 0.05$ compared with WT mice, #$P < 0.05$ compared with *CAV1⁺/⁻* mice. **d** Relative RV weight was determined as the ratio of the RV weight to LV and septum weights (RV/LV + S). ****$P < 0.0001$ compared with WT mice, #$P < 0.05$ compared with *CAV1⁺/⁻* mice. **e** Pulmonary vascular remodeling was assessed by measuring the medial thickness of distal pulmonary vessels from lung sections immunostained with anti-αSMA antibody. Percent wall thickness is expressed as the medial wall area divided by the area of the vessel. **$P < 0.01$ compared with WT mice, #$P < 0.05$ compared with *CAV1⁺/⁻* mice. Scale bar, 20 μm. **f** Phosphorylation of Smad 1/5/9 was evaluated in PMVECs isolated from WT mice, *CAV1⁺/⁻* mice, and *CAV1⁺/⁻/Cavin-1⁺/⁻* mice. *$P < 0.05$ compared with WT mice, #$P < 0.05$ compared with *CAV1⁺/⁻* mice. Data are expressed as the mean ± standard error.

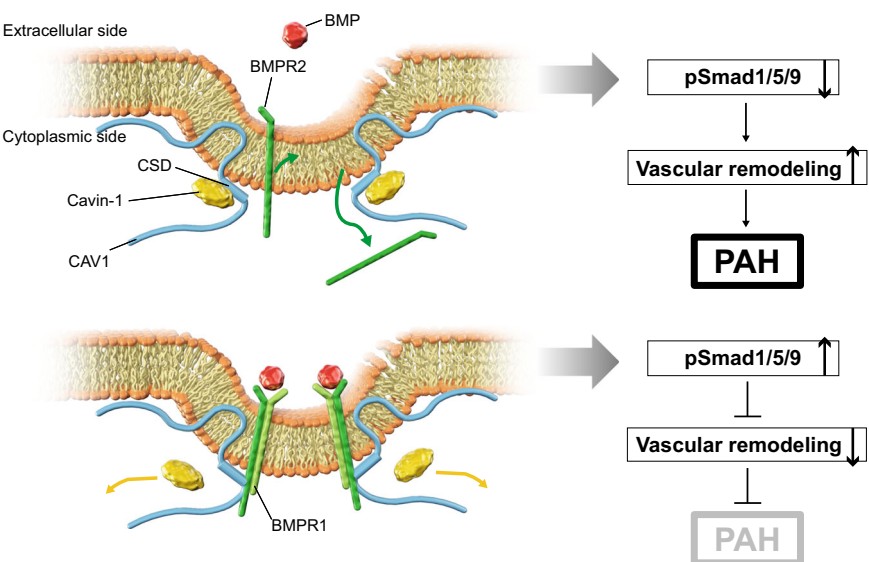

**Fig. 7 Working hypothesis.** Proposed roles of Cavin-1 include competitively inhibiting the binding of CAV1 and BMPR2 in the caveolin scaffolding domain and attenuating BMPR2 localization at the plasma membrane and BMP/Smad signaling in the progression of PAH. CSD caveolin scaffolding domain.

described[17]. Briefly, GST-CAV1(FL), GST-CAV1(1–61), GST-CAV1(61–101), or GST-CAV1(135–178) proteins were bound to glutathione-Sepharose beads. The beads were incubated with lysates from 293FT cells transfected with the FLAG-tagged Cavin-1 or myc-tagged BMPR2 expression plasmids at 4 °C for 2 h. GST-conjugated glutathione-Sepharose beads were used as a control. The beads were washed five times and the proteins were eluted for western blotting.

**BioID screening**. The Humanized BirA (hBirA) sequence was obtained from a previously published sequence[73] by adding a Kozac sequence at the 5' site and an Eco-RI sequence at the 3' site. This sequence was inserted into pcDNA3.1(+)-HA (Clonetech, CA, USA) to generate pcDNA3.1 + /hBirA-HA. HA-tagged hBirA was cloned by PCR from the pcDNA3.1 + /hBirA-HA plasmid with forward and reverse primers (F: 5'-TTAGATCTCTCGAGGCC ACCATGAAGGACAACACCGTGCC-3',ʹR: 5'- CCTACCCGGTA GAATTCTCAAGCGTAATCTGGAACATCGTATGGGTACTTC TCTGCGCTTCTCAGGG-3'). HA-tagged hBirA was inserted into pMSCV-puromycin to generate pMSCV-hBirA-HA-puromycin using the Gibson Assembly® kit (New England Biolabs, Ipswich, MA, USA). HA-tagged hCAV1 was cloned by PCR from the pcDNA3.1-T7-hCAV1 plasmid with forward and reverse primers (F: 5'-GAAGCGCAGAGAAGGAATTCATGTCTGGGGGCAAA-TACGT-3,' R: 5'-CCTACCCGGTAGAATTCTCAAGCGTAATCT GGAACATCGTATGGGTATATTTCTTTCTGCAAGTTGA-3'). HA-tagged hCAV1 was inserted into pMSCV-hBirA-HA-puromycin to create pMSCV-hBirA-hCAV1-HA-puromycin.

hPAECs expressing hBirA-HA or hBirA-hCAV1-HA by retrovirus infection were incubated with 50 μM biotin for 6 h. The lysate was collected and mixed with MagCapture Tamavidin2-REV beads (FUJIFILM Wako Pure Chemical Corporation, Tokyo, Japan) at 4 °C overnight. The beads were washed with PBS, and mass spectrometry was performed (CERI, Tokyo, Japan) as previously described[74]. LC–MS/MS analyses were performed using a nano LC (UltiMate® 3000) (Dionex, Sunnyvale, CA, USA) coupled with a Q Exactive Plus Orbitrap mass spectrometer (Thermo Scientific, Waltham, MA, USA). Instrument operation and data acquisition were performed using Xcalibur Software (Thermo Scientific, Waltham, MA, USA).

**LC–MS/MS data analysis**. The MS/MS raw data were processed using Mascot version 2.6.0 (Matrix Sciences, London, US) and were searched in the Swiss-Prot database with humans as the species, carbamidomethylation of cysteine as a static modification, oxidation of methionine as a dynamic modification, precursor mass tolerance of 1.0 Da, and a fragment mass tolerance of 0.8 Da. The dat files of all fractions obtained were processed with Scaffold version 5.0.1 (Proteome Software Inc.). The parameters of scaffold were adjusted as follows to ensure accurate identification of peptides and proteins: protein identifications were accepted if they were established with a probability greater than 90% and peptide identifications were accepted if they contained at least one peptide identified with a probability greater than 95% (using the Scaffold Local FDR algorithm). The normalized spectral abundance factor (Weighted Spectra) was calculated for each

protein and compared between CAV1 and HA samples. Proteins with weighted spectra greater than 10 were considered significant, and differences were analyzed with Fisher's exact test using Scaffold. $P < 0.01$ was considered statistically significant.

**Statistics and reproducibility**. The experiments were performed at least three times. All data are expressed as the mean ± standard error. The numerical source data underlying the graphs from this study is present in the supplementary data file and that uncropped blots can be found in Supplementary Fig. 10. Statistical analysis was performed using one-way ANOVA followed by Tukey's post-hoc test. Statistical significance was set at $P < 0.05$. Statistical analyses were performed using GraphPad Prism 8 (GraphPad Software, Inc., CA, USA).

**Reporting summary**. Further information on research design is available in the Nature Portfolio Reporting Summary linked to this article.

## Data availability

The authors declare that all data supporting the findings of this study are available within the article and its Supplementary Information Files. The source data behind the BioID screening is available in Supplementary Data 1. Any remaining information can be obtained from the corresponding author upon reasonable request.

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

## Acknowledgements

This work was supported by the Japan Society for the Promotion of Science (JSPS) KAKENHI Grant Number 18K08111, by a grant-in-aid from the Public Promoting Association Asano Foundation for Studies on Medicine, and by a grant from Actelion Academic Prize. We would like to acknowledge Editage (www.editage.com) for providing assistance with the English language editing.

## Author contributions

S.T. and N.N. planned and performed most of the experiments and participated in manuscript writing. T.S. and Y.T. performed in vitro experiments. Y.H. and A.S. performed in vivo experiments. T.O. conceived and designed the study, provided funding for the research, and participated in manuscript writing. S.M. supervised the study and reviewed the manuscript. N.N. designed and directed the study, provided funding for the research and wrote the manuscript.

## Competing interests

The authors declare no competing interests.
