## [Peer Review File · Communications Biology]

Reviewers' comments:

Reviewer #1 (Remarks to the Author):

This is an interesting manuscript describing a role for CAV1-Cavin-1 interaction in the regulation of BMP/SMAD signaling in pulmonary arterial hypertension. Using isolated pulmonary microvascular endothelial cells and mouse models they show that expression of cavin-1, an adaptor protein required for caveolae formation by caveolin-1, regulates CAV1 control of BMP/SMAD signaling. Overall an interesting and well-performed manuscript.

Comments

1. The idea that CAVIN-1 competes with the CSD for BMPR-2 is interesting and the PLA data provided very convincing. A role for the CSD should be shown more directly using CSD mutant constructs for both the PLA and coIP expts. It would also be important to extend the coIP expts to IP CAV1 and show interaction between CAV1 and either CAVIN-1 or BMPR2 as well as see the effect of hypoxia on complex interaction. Demonstration of direct competition would require in vitro assays, perhaps using the GST constructs used in Fig 4D.

2. An alternative hypothesis is that BMPR-2 is interacting not with caveolae but with non-caveolar scaffolds and that CAVIN-1 induction of caveolae is reducing the pool of available scaffolds for interaction. This, and more generally the expression and established role of non-caveolar Cav1 domains in signaling, should be considered and discussed. Exclusion of this hypothesis seems to be based on the heterozygote mouse data in Figure 6 where CAVIN-1 knockdown does not significantly reduce the number of caveolae. However this data shows clearly a reduction in caveolae and increasing sample numbers would be critical to determine if this reduction is truly not significant.

3. The use of "membrane" to describe the post nuclear supernatant is incorrect. Also the two-step fractionation approach may result in membrane contamination of the "cytosol" fraction and this should be tested by probing these fraction for ER markers.

4. On page 11, Supp Fig 6 should be Supp Fig 7. The Western blot data in Supp Fig 7 is critical to understanding the data and should be included in Figure 6 (perhaps without the quantification to save space)

Reviewer #2 (Remarks to the Author):

The authors show that BMP/Smad signaling is suppressed in pulmonary microvascular endothelial cells of CAV1 knockout mice. Hypoxia enhanced the CAV1/Cavin-1 interaction but attenuated the CAV1/BMPR2 interaction and BMPR2 membrane localization in pulmonary artery endothelial cells. Cavin-1 decreased BMPR2 membrane localization by inhibiting the interaction of BMPR2 with CAV1 and reduced Smad signal transduction. However, the authors fail to cite crucial literature. For example Jiang et al in 2010 showed already the association and misdirection of BMPR2 on the plasma membrane and how it effects Smad signaling. The data obtained here need to be brought into context with previous literature. Also the interaction of BMP receptors with caveolae is well described in the literature and the shuttling of BMP receptors for activating signaling. Moreover the location of BMP receptors in caveolae was found to be important for their signaling (akkiraju et al., 2011) etc.

Based on the histology the authors conclude that CAV1 deficiency led to downregulation of Smad 1/5/9 signaling without BMPR2 reduction in PAECs, however they do not show if it is direct or indirect effect.

In Figure 2 it looks like one of the siRNA affect pSmad2. In figure 2 the cells not treated with siRNA should be included as a control to show if there are differences in the treatment with siRNA. The authors should show if BMPR2 is present in the golgi or ER or in endosomes when cavin-1 is decreased.

The size of BMPR2 is usually about 180kd. The BMPR2 without the cytoplasmic tail that is a spliced variant is about 100kd. In the Western blots are the authors detecting the splice variant?

The authors should include some z scans to demonstrate the localization of BMPRIa on the plasma membrane and interaction with cavin1 and cav1.

In Figure 5 caveolae and cavin are proteins that are also described to be present in vesicles in the cytosol. However they are not detected at all in the Western blot. This suggests there is some issue with the blots. In order to be secreted to the plasma membrane every protein has to pass through the RE and Golgi. Are these included in the membrane fraction?

Reviewer #3 (Remarks to the Author):

COMMSBIO-23-0447-T The Authors of the manuscript entitled "The Cavin-1/Caveolin-1 interaction attenuates BMP/Smad signaling in pulmonary hypertension by interfering with BMPR2/Caveolin-1 binding" have shown the role and interrelationship among caveolin-1, cavin-1 and BMPR2 signaling in pulmonary hypertension using CAV1-knockout (CAV1^{-/-}) mice (C57BL/6J background) and Cavin-1-knockout (Cavin-1^{-/-}) mice (C57BL/6J background). It is an interesting study.

“The Cavin-1/Caveolin-1 interaction attenuates BMP/Smad signaling in pulmonary hypertension by interfering with BMPR2/Caveolin-1 binding”

(Manuscript #: COMMSBIO-23-0447-T)

Shinya Tomita, Naohiko Nakanishi, Takehiro Ogata, Yusuke Higuchi, Akira Sakamoto, Yumika Tsuji, Takaomi Suga, Satoaki Matoba

We thank the Editor and Reviewers for their interest and constructive suggestions in our manuscript. As advised by the Editor and Reviewers, we have responded to their comment and suggestions, and we hope that we have adequately modified the revised manuscript to address their concerns.

Responses to all queries are highlighted with Track Changes in the revised manuscript.

Reviewer 1

This is an interesting manuscript describing a role for CAV1-Cavin-1 interaction in the regulation of BMP/SMAD signaling in pulmonary arterial hypertension. Using isolated pulmonary microvascular endothelia cells and mouse models they show that expression of cavin-1, an adaptor protein required for caveolae formation by caveolin-1, regulates CAV1 control of BMP/SMAD signaling. Overall, an interesting and well-performed manuscript.

1. The idea that CAVIN-1 competes with the CSD for BMPR-2 is interesting and the PLA data provided very convincing. A role for the CSD should be shown more directly using CSD mutant constructs for both the PLA and coIP expts. It would also be important to extend the coIP expts to IP CAV1 and show interaction between CAV1 and either CAVIN-1 or BMPR2 as well as see the effect of hypoxia on complex interaction. Demonstration of direct competition would require in vitro assays, perhaps using the GST constructs used in Fig 4D.

Response:

We very much appreciate your valuable comments. To more directly evaluate the competitive binding of CAV1 CSD to Cavin-1 and BMPR2, we examined the binding of CAV1 Δ CSD (81-101 deletion) to Cavin-1 and BMPR2 using co-immunoprecipitation, and competitive association of Cavin-1 and BMPR2 with the fragment of CAV1 (61-101) using GST pulldown assay. In immunoprecipitation, full-length CAV1 was associated with Cavin-

1 and BMPR2; however, CAV1 Δ CSD did not associate with both Cavin-1 and BMPR2, suggesting that both Cavin-1 and BMPR2 are associated with CAV1 CSD domain (Figure 4E). In the GST pulldown assay, Cav1(61–101), which is an oligomerization domain containing a CSD in the distal half, was associated with both Cavin-1 and BMPR2. The association of CAV1 (61-101) with Cavin-1 was decreased by co-expression of BMPR2, and as well as Cavin-1, the association of CAV1 (61-101) with BMPR2 was reduced by co-expression of Cavin-1 (Figure 4F). These results suggest that Cavin-1 and BMPR2 are competitively associated with the CAV1 CSD domain.

We added these results in Figure 4E and 4F and revised the Result section in our manuscript as follow.

(p10, line 21–p11, line 4)

‘Full-length CAV1 was associated with Cavin-1 and BMPR2 in immunoprecipitation; however, CAV1 Δ CSD did not associate with both Cavin-1 and BMPR2 (Figure 4E), suggesting that both Cavin-1 and BMPR2 are associated with CAV1 CSD domain. In addition, the association of CAV1 (61-101) with Cavin-1 was decreased by co-expression of BMPR2, and as well as Cavin-1, the association of CAV1 (61-101) with BMPR2 was reduced by co-expression of Cavin-1 (Figure 4F).’

E**F**
2. An alternative hypothesis is that BMPR-2 is interacting not with caveolae but with non-caveolar scaffolds and that CAVIN-1 induction of caveolae is

reducing the pool of available scaffolds for interaction. This, and more generally the expression and established role of non-caveolar Cav1 domains in signaling, should be considered and discussed. Exclusion of this hypothesis seems to be based on the heterozygote mouse data in Figure 6 where CAVIN-1 knockdown does not significantly reduce the number of caveolae. However, this data shows clearly a reduction in caveolae and increasing sample numbers would be critical to determine if this reduction is truly not significant.

Response:

Thank you for your comments. As you mentioned, we also need to consider the function of CAV1 in non-caveolae and the interaction of BMP2 with non-caveolar CAV1. In general, caveolins form caveolae together with cavins. However, several cells, such as hepatocytes, neurons, and lymphocytes, express CAV1 without forming distinguishable caveolae. Non-caveolar caveolins also organize scaffolds that interact with proteins and play the roles of lipids transportation or signal transduction¹. In addition, it has been reported that non-caveolar CAV1 expression promotes lymphangiogenesis², and Cavin-1 neutralizes non-caveolar CAV1 microdomains in prostate cancer³. Therefore, we agree that your alternative hypothesis is reasonable. To address your hypothesis, we re-investigated whether the number of caveolae was decreased or not (not only classical flask-shaped invagination but also bulb-shaped pits near plasma membrane) in WT, *CAV1*^{+/-}, and *CAV1*^{+/-}/*Cavin-1*^{+/-} mice. As described below, the number of caveolae in PAECs of *CAV1*^{+/-}/*Cavin-1*^{+/-} mice was not

decreased compared with that of *CAV1*^{+/-} mice. This result suggests, but insufficiently, that Cavin-1 does not affect the scaffolding of non-caveolar CAV1.

The localization of BMP receptors is also essential for BMP signaling. A part of BMPR2 mutants misdirect the localization of BMP receptors on the plasma membrane and decrease BMP-dependent Smad signaling⁴. In addition, the localization of BMP receptors in caveolae is important for activating Smad signaling⁵. In the present study, Cavin-1 decreased BMPR2 localization on the plasma membrane, and the knockdown of Cavin-1 rescued its localization to the plasma membrane and the Smad 1/5/9 phosphorylation which were reduced by CAV1 knockdown. These findings suggest that Cavin-1 regulates the Smad signaling through the BMPR2 localization on the plasma membrane.

We added these discussions and references in the Discussion section and revised our manuscript as follows.

(p14, line 15–p15, line 12)

‘In general, caveolins form caveolae together with cavins. However, several cells, such as hepatocytes, neurons, and lymphocytes, express CAV1 without forming distinguishable caveolae. Non-caveolar caveolins also organize scaffolds that interact with proteins and play the roles of lipids transportation or signal transduction³⁴. In addition, it has been reported that non-caveolar CAV1 expression promotes lymphangiogenesis³⁵, and Cavin-1 neutralizes non-caveolar CAV1 microdomains in prostate cancer³⁶.

Therefore, an alternative hypothesis is that BMPR2 interacts not with caveolae but with non-caveolar scaffolds and that Cavin-1 induction of caveolae reduces the pool of available scaffolds for interaction. However, the number of caveolae in PAECs of *CAV1^{+/-}/Cavin-1^{+/-}* mice was not decreased compared with that of *CAV1^{+/-}* mice. This result suggests, but insufficiently, that Cavin-1 does not affect the scaffolding of non-caveolar CAV1. The localization of BMP receptors is also essential for BMP signaling. A part of BMPR2 mutants misdirect the localization of BMP receptors on the plasma membrane and decrease BMP-dependent Smad signaling³⁷. In addition, the localization of BMP receptors in caveolae are important for activating Smad signaling³⁸. In the present study, Cavin-1 decreased BMPR2 localization on the plasma membrane, and the knockdown of Cavin-1 rescued its localization to the plasma membrane and the Smad 1/5/9 phosphorylation which were reduced by CAV1 knockdown. These findings suggest that Cavin-1 regulates the Smad signaling through the BMPR2 localization on the plasma membrane.'

1. Pol A, Morales-Paytuvi F, Bosch M, Parton RG. Non-caveolar caveolins - duties outside the caves. *J Cell Sci* 133, (2020).
2. Nassar ZD, Hill MM, Parton RG, Francois M, Parat MO. Non-caveolar caveolin-1 expression in prostate cancer cells promotes lymphangiogenesis. *Oncoscience* 2, 635-645 (2015).
3. Moon H, *et al.* PTRF/cavin-1 neutralizes non-caveolar caveolin-1 microdomains in prostate cancer. *Oncogene* 33, 3561-3570 (2014).

4. Jiang Y, et al. Trapping of BMP receptors in distinct membrane domains inhibits their function in pulmonary arterial hypertension. *Am J Physiol Lung Cell Mol Physiol* 301, L218-227 (2011).
5. Moseychuk O, et al. Inhibition of CK2 binding to BMPRIa induces C2C12 differentiation into osteoblasts and adipocytes. *J Cell Commun Signal* 7, 265-278 (2013).

3. The use of "membrane" to describe the post nuclear supernatant is incorrect. Also, the two-step fractionation approach may result in membrane contamination of the "cytosol" fraction and this should be tested by probing these fractions for ER markers.

Response:

Thank you for pointing out some critical issues. We agree with your comments. The membrane fraction by the two-step fractionation method we performed, contained the plasma membrane and several organelles: GM130 as a Golgi marker, PDI as an ER marker, and Rab7 as an endosome marker were included in the membrane fraction as well as VE-cadherin and CAV1. Therefore, the membrane fraction in the present study contained plasma membrane and these organelles.

We added the Western blotting probing the organelles in the membrane fraction as Supplementary Figure 4A and revised our manuscript as follows.

(p7, lines 17–22)

‘The membrane fraction by the two-step fractionation method we performed, contained the plasma membrane and several organelles: GM130 as a Golgi marker, PDI as an ER marker, and Rab7 as an endosome marker were included in the membrane fraction as well as VE-cadherin and CAV1 (Supplementary Figure 4A). Therefore, the membrane fraction in the present study contained plasma membrane and these organelles.’

A

4. On page 11, Supp Fig 6 should be Supp Fig 7. The Western blot data in Supp Fig 7 is critical to understanding the data and should be included in Figure 6 (perhaps without the quantification to save space)

Response:

We apologize for our mistake. As you point out, these data are very important. We moved these Western blot images from Supplementary Figure to Figure 6A.

Reviewer 2

1. The authors show that BMP/Smad signaling is suppressed in pulmonary microvascular endothelial cells of CAV1 knockout mice. Hypoxia enhanced the CAV1/Cavin-1 interaction but attenuated the CAV1/BMPR2 interaction and BMPR2 membrane localization in pulmonary artery endothelial cells. Cavin-1 decreased BMPR2 membrane localization by inhibiting the interaction of BMPR2 with CAV1 and reduced Smad signal transduction. However, the authors fail to cite crucial literature. For example, Jiang et al in 2010 showed already the association and misdirection of BMPR2 on the plasma membrane and how it effects Smad signaling. The data obtained here need to be brought into context with previous literature. Also, the interaction of BMP receptors with caveolae is well described in the literature and the shuttling of BMP receptors for activating signaling. Moreover, the location of BMP receptors in caveolae was found to be important for their signaling (akkiraju et al., 2011) etc.

Response:

We very much appreciate your valuable comments. As you pointed out, it has been reported that the localization of BMP receptors is essential for their signaling. A part of BMPR2 mutants misdirect the localization of BMP receptors on the plasma membrane and decrease BMP-dependent Smad signaling⁴. In addition, the localization of BMP receptors in caveolae are important for activating Smad signaling⁵. In the present study, Cavin-1 decreased BMPR2 localization on the plasma membrane, and the

knockdown of Cavin-1 rescued its localization to the plasma membrane and the Smad 1/5/9 phosphorylation which were reduced by CAV1 knockdown. These findings suggest that Cavin-1 regulates the Smad signaling through the BMPR2 localization on the plasma membrane.

We added these discussions and references in the Discussion section in the revised manuscript as follows.

(p15, lines 3–12)

‘The localization of BMP receptors is also essential for BMP signaling. A part of BMPR2 mutants misdirect the localization of BMP receptors on the plasma membrane and decrease BMP-dependent Smad signaling³⁷. In addition, the localization of BMP receptors in caveolae are important for activating Smad signaling³⁸. In the present study, Cavin-1 decreased BMPR2 localization on the plasma membrane, and the knockdown of Cavin-1 rescued its localization to the plasma membrane and the Smad 1/5/9 phosphorylation which were reduced by CAV1 knockdown. These findings suggest that Cavin-1 regulates the Smad signaling through the BMPR2 localization on the plasma membrane.’

4. Jiang Y, *et al.* Trapping of BMP receptors in distinct membrane domains inhibits their function in pulmonary arterial hypertension. *Am J Physiol Lung Cell Mol Physiol* 301, L218-227 (2011).
5. Moseychuk O, *et al.* Inhibition of CK2 binding to BMPRIa induces C2C12 differentiation into osteoblasts and adipocytes. *J Cell Commun Signal* 7, 265-278 (2013).

2. Based on the histology the authors conclude that CAV1 deficiency led to downregulation of Smad 1/5/9 signaling without BMPR2 reduction in PAECs, however they do not show if it is direct or indirect effect.

Response:

We appreciate your constructive comments. We demonstrated that total BMPR2 expression was not reduced in *CAV1*^{-/-} PMVECs and PSMCs, but Smad 1/5/9 phosphorylation was significantly decreased in *CAV1*^{-/-} PMVECs compared to WT PMVECs (Figure 1B). We also confirmed that Smad2 phosphorylation, which is downstream of TGF signaling, was not decreased in *CAV1*^{-/-} PMVECs (Figure 1B). In vitro study, *CAV1* knockdown reduced membranous localization of BMPR2 without reduction in total protein level, and decreased Smad 1/5/9 phosphorylation (Figure 2). As mentioned in Concern 1, the localization of BMP receptors is important for their signaling. Integrating the results of histology and molecular biological analyses, our findings suggest that that BMPR2 reduction at caveolae in *CAV1*^{-/-} PMVECs directly decrease Smad 1/5/9 phosphorylation.

3. In Figure 2 it looks like one of the siRNA affect pSmad2. In figure 2 the cells not treated with siRNA should be included as a control to show if there are differences in the treatment with siRNA.

Response:

Thank you for your comments. To evaluate the effect of siRNA, WB was performed using hPAECs without siRNA treatment (naive PAECs). pSmad2 was rarely detected under no stimulation. BMPR2, pSmad1/5/9 and pSmad2 were not different between naive and control siRNA-transfected PAECs. Therefore, we found that siRNA did not affect pSmad2. According to your suggestions, we revised Figure 2A to contain naive PAECs.

4. The authors should show if BMPR2 is present in the golgi or ER or in endosomes when cavin-1 is decreased.

Response:

Thank you for your comments. As you mentioned, the localization of BMPR2 in CAV1-knockdown PAECs is an important issue. Additional immunostaining (Supplementary Figure 4B, 4C, and 4D) showed that BMPR2 was partially co-localized with PDI as an ER marker, GM130 as a Golgi marker, or Rab7 as an endosome marker in PAECs. Co-localization of BMPR2 with PDI, GM130, and Rab7 was not apparently changed by CAV1 knockdown in PAECs. Considering the results of Western blotting that these organelles were detected in the membrane fraction and BMPR2 was increased in the cytosol fraction in CAV1 knockdown PAECs, our results suggest that CAV1 knockdown promotes BMPR2 translocation from the membrane to the cytoplasm in PAECs.

We added the Western blotting probing these organelles in membrane fraction as Supplementary Figure 4A and the immunostaining images of BMPR2 and organelles in CAV1 knockdown PAECs as Supplementary Figure 4B, C, and D, and revised our manuscript as follows.

(p8, line 5–line 12)

‘Immunostaining showed that BMPR2 was partially co-localized with PDI, GM130, and Rab7 in PAECs. Co-localization of BMPR2 with PDI, GM130, and Rab7 was not apparently changed by CAV1 knockdown in PAECs (Supplementary Figure 4B, 4C, and 4D). Considering the results of Western blotting that these organelles were detected in the membrane fraction and BMPR2 was increased in the cytosol fraction in CAV1 knockdown PAECs,

our results suggest that CAV1 knockdown promotes BMPR2 translocation from the membrane to the cytoplasm in PAECs.'

5. The size of BMPR2 is usually about 180kd. The BMPR2 without the cytoplasmic tail that is a spliced variant is an about 100kd. In the Western blots are the authors detecting the splice variant?

Response:

We appreciate your comments. We used a commercially available antibody against BMPR2 (3F6F8, MA5-15827, Thermo Fisher Scientific Inc.). In the datasheet of this antibody, ~115kDa band corresponding to BMPR2 is observed across cell lines tested⁶. As you mentioned, these bands might be detecting the splice variant of BMPR2 by Western blotting. We added the following reference regarding the use of the antibody in the revised manuscript.

6. **Qian S, *et al.* BMPR2 promotes fatty acid oxidation and protects white adipocytes from cell death in mice. *Commun Biol* 3, 200 (2020).**

6. The authors should include some z scans to demonstrate the localization of BMPR1a on the plasma membrane and interaction with cavin1 and cav1.

Response:

Thank you for your comments. According to your suggestion, we examined BMPR1a localization in CAV1-knockdown PAECs. Same as BMPR2, BMPR1a at the plasma membrane was also decreased in CAV1-knockdown PAECs. We added this result in Supplementary Figure 5 and revised our manuscript as follows.

(p8, lines 12–13)

‘In addition, BMPR1a at the plasma membrane was also decreased in CAV1-knockdown PAECs (Supplementary Figure 5).’

Supplementary Figure 5. Representative immunostaining images of Cavin-1 and BMPR1a in CAV1-knocked down PAECs.
Representative fluorescent images of Cavin-1 and BMPR1a in control and CAV1-knocked down hPAECs. Scale bar, 50 μ m.

7. In Figure 5 caveolae and cavin are proteins that are also described to be present in vesicles in the cytosol. However, they are not detected at all in the Western blot. This suggests there is some issue with the blots. In order to be secreted to the plasma membrane every protein has to pass through the RE and Golgi. Are these included in the membrane fraction?

Response:

Thank you for pointing out an important issue. The membrane fraction by the two-step fractionation method we performed, contained the plasma membrane and several organelles: GM130 as a Golgi marker, PDI as an ER marker, and Rab7 as an endosome marker were included in the membrane fraction as well as VE-cadherin and CAV1. Therefore, the membrane fraction in the present study contained plasma membrane and these organelles.

We added the Western blotting probing these organelles in membrane fraction as Supplementary Figure 4A and revised our manuscript as follows.

(p7, lines 17–22)

‘The membrane fraction by the two-step fractionation method we performed, contained the plasma membrane and several organelles: GM130 as a Golgi marker, PDI as an ER marker, and Rab7 as an endosome marker were included in the membrane fraction as well as VE-cadherin and CAV1. (Supplementary Figure 4A). Therefore, the membrane fraction in the present study contained plasma membrane and these organelles.’

A

Reviewer 3

COMMSBIO-23-0447-T The Authors of the manuscript entitled “The Cavin-1/Caveolin-1 interaction attenuates BMP/Smad signaling in pulmonary hypertension by interfering with BMPR2/Caveolin-1 binding” have shown the role and interrelationship among caveolin-1, cavin-1 and BMPR2 signaling in pulmonary hypertension using CAV1-knockout (CAV1^{-/-}) mice (C57BL/6J background) and Cavin-1-knockout (Cavin-1^{-/-}) mice (C57BL/6J background). It is an interesting study.

Response:

We greatly appreciate the time and effort you have dedicated to providing insightful feedback. We believe our results provide insights into pulmonary hypertension and contribute to developing novel therapies to enable reverse remodeling of the pulmonary artery. Thank you again for your cooperation.

Reviewers' comments:

Reviewer #1 (Remarks to the Author):

This remains a very interesting story that describes cavin-1 - caveolin-1 regulation of BMPR-2 signaling to pSMAD1,5,9 in cardiac cells. Of particular interest is the idea that Cav1-cavin-1 stoichiometry is key to regulation of Cav1 signaling function of BMPR-2 to pSMAD1,5,9. Reduced membrane associated BMPR2 is restored in CAV1 knockdown cells by knockdown of cavin-1. I accept the author's argument that the fact that cavin-1 knockdown does not alter the number of caveolae that the impact on signaling is not due to changes in the relative abundance of caveola vs scaffolds.

This lack of an effect on caveolae number does however raise questions as to whether the release of cavin-1 that promotes BMPR-2 recruitment to the membrane is actually from caveolae. The authors do address this point in the discussion but should correctly cite the literature. The first demonstration of a role for non-caveolar Cav1 scaffolds in signaling was PMID: 17938246 and the term scaffolds coined in PMID: 19398762. These papers should be cited. They should also note that inhibition of Cav1 signaling function by expression of cavin-1 has been shown previously (PMID: 25942420 PMID: 20732728).

The authors argue that the underlying mechanism is competition between BMPR-2 and cavin1 for the Cav1 CSD. The data showing that BMPR-2 and cavin-1 bind to the CSD is convincing but that for competition less so, essentially based on figure 4F. This figure shows that expression of cavin-1 reduces (slightly) BMPR-2 association with Cav1 61-101-GST. This figure isn't strong as it is based on over expression studies in cells. Also there is a background anti-myc band that comigrates with BMPR-2-myc even in cells that were not transfected with BMPR-2 myc. Finally the huge increase in the BMPR-2-myc band when co-transfected with cavin-1 is confounding. I have two suggestions to further support this conclusion:

1. a more direct experiment to show competition would be in vitro pulldowns using purified cavin-1 to compete off BMPR-2
2. The authors include experiments showing that overexpression of Cavin-1 displaces BMPR-2 from the membrane. Does cavin-1 overexpression in WT cells also reduce pSMAD1.5.9 signaling?

The non-specific spots in the Cav1 images of Figure 6B raise questions as to specificity of the labeling and these experiments should be repeated with improved labeling.

Reviewer #2 (Remarks to the Author):

The authors addressed the critiques raised

“The Cavin-1/Caveolin-1 interaction attenuates BMP/Smad signaling in pulmonary hypertension by interfering with BMPR2/Caveolin-1 binding”

(Manuscript #: COMMSBIO-23-0447A)

Shinya Tomita, Naohiko Nakanishi, Takehiro Ogata, Yusuke Higuchi, Akira Sakamoto, Yumika Tsuji, Takaomi Suga, Satoaki Matoba

We thank the Editor and Reviewers for their interest and constructive suggestions in our manuscript. As advised by the Editor and Reviewers, we have responded to their comment and suggestions, and we hope that we have adequately modified the revised manuscript to address their concerns.

Responses to all queries are highlighted in the revised manuscript.

Reviewer 1

This remains a very interesting story that describes cavin-1 - caveolin-1 regulation of BMPR-2 signaling to pSMAD1,5,9 in cardiac cells. Of particular interest is the idea that Cav1-cavin-1 stoichiometry is key to regulation of Cav1 signaling function of BMPR-2 to pSMAD1,5,9. Reduced membrane associated BMPR2 is restored in CAV1 knockdown cells by knockdown of cavin-1. I accept the author's argument that the fact that cavin-1 knockdown does not alter the number of caveolae that the impact on signaling is not due to changes in the relative abundance of caveola vs scaffolds.

1. This lack of an effect on caveolae number does however raise questions as to whether the release of cavin-1 that promotes BMPR-2 recruitment to the membrane is actually from caveolae. The authors do address this point in the discussion but should correctly cite the literature. The first demonstration of a role for non-caveolar Cav1 scaffolds in signaling was PMID: 17938246 and the term scaffolds coined in PMID: 19398762. These papers should be cited. They should also note that inhibition of Cav1 signaling function by expression of cavin-1 has been shown previously (PMID: 25942420 PMID: 20732728).

Response:

We very much appreciate your valuable comments. We cited these papers in the discussion section.

(p14, lines 19–24)

'Non-caveolar caveolins also organize scaffolds that interact with proteins and play the roles of lipids transportation or signal transduction³⁴⁻³⁶. In addition, it has been reported that non-caveolar CAV1 expression promotes lymphangiogenesis³⁷, and Cavin-1 neutralizes non-caveolar CAV1 microdomains and inhibits CAV1 signaling function in prostate cancer³⁸⁻⁴⁰.'

34. Lajoie P, et al. Plasma membrane domain organization regulates EGFR signaling in tumor cells. *J Cell Biol* 179, 341-356 (2007). 10.1083/jcb.200611106, Pubmed:17938246.

35. Lajoie P, Goetz JG, Dennis JW, Nabi IR. Lattices, rafts, and scaffolds: domain regulation of receptor signaling at the plasma membrane. *J Cell Biol* 185, 381-385 (2009). 10.1083/jcb.200811059, Pubmed:19398762.

38. Meng F, Joshi B, Nabi IR. Galectin-3 Overrides PTRF/Cavin-1 Reduction of PC3 Prostate Cancer Cell Migration. *PLoS One* 10, e0126056 (2015). 10.1371/journal.pone.0126056, Pubmed:25942420.

39. Aung CS, Hill MM, Bastiani M, Parton RG, Parat MO. PTRF-cavin-1 expression decreases the migration of PC3 prostate cancer cells: role of matrix metalloprotease 9. *Eur J Cell Biol* 90, 136-142 (2011). 10.1016/j.ejcb.2010.06.004, Pubmed:20732728.

2. The authors argue that the underlying mechanism is competition between BMPR-2 and cavin1 for the Cav1 CSD. The data showing that BMPR-2 and cavin-1 bind to the CSD is convincing but that for competition less so, essentially based on figure 4F. This figure shows that expression of cavin-1 reduces (slightly) BMPR-2 association with Cav1 61-101-GST. This figure isn't strong as it is based on over expression studies in cells. Also there is a

background anti-myc band that comigrates with BMPR-2-myc even in cells that were not transfected with BMPR-2 myc. Finally the huge increase in the BMPR-2-myc band when co-transfected with cavin-1 is confounding. I have two suggestions to further support this conclusion:

1. a more direct experiment to show competition would be in vitro pulldowns using purified cavin-1 to compete off BMPR-2
2. The authors include experiments showing that overexpression of Cavin-1 displaces BMPR-2 from the membrane. Does cavin-1 overexpression in WT cells also reduce pSMAD1.5.9 signaling?

Response:

We very much appreciate your comments and suggestions. We attempted to overexpress Cavin-1 using plasmid transfection or retrovirus infection but unfortunately could not obtain sufficient overexpression of Cavin-1 in hPAECs to evaluate Smad phosphorylation in Western blotting.

According to your suggestion, we performed pulldown assay using recombinant Cavin-1 protein in vitro. Consistent with other results, Cavin-1 was found to decrease the binding of BMPR2 to CAV1 (61-101). This result strengthens our hypothesis that Cavin-1 and BMPR2 are competitively associated with the CAV1 CSD domain.

We added this result in Figure 4G and revised the Result section in our manuscript as follows.

(p11, lines 4–5)

‘Consistent with these results, recombinant Cavin-1 protein was found to decrease the binding of BMPR2 to CAV1 (61-101) (Figure 4G).’

G

3. The non-specific spots in the Cav1 images of Figure 6B raise questions as to specificity of the labeling and these experiments should be repeated with improved labeling.

Response:

We agree your concern. We re-evaluated the immunostaining images in Figure 5B and confirmed that Cavin-1 knockdown improved BMPR2 localization which was reduced by CAV1 knockdown. We changed the Figure 5B as follow. We appreciate your suggestions for improving the figure.

B

REVIEWERS' COMMENTS:

Reviewer #1 (Remarks to the Author):

All concerns addressed. Accept.